# Active Ranking without Strong Stochastic Transitivity

**Hao Lou**
Dept. of Electrical & Computer Engineering
University of Virginia
Charlottesville, VA 22903
haolou@virginia.edu

**Tao Jin**
Department of Computer Science
University of Virginia
Charlottesville, VA 22903
taoj@virginia.edu

**Yue Wu**
Department of Computer Science
University of California, Los Angeles
Los Angeles, CA 90095
ywu@cs.ucla.edu

**Pan Xu**
Dept. of Biostatistics & Bioinformatics
Duke University
Durham, NC 27705
pan.xu@duke.edu

**Quanquan Gu**[*]
Department of Computer Science
University of California, Los Angeles
Los Angeles, CA 90095
qgu@cs.ucla.edu

**Farzad Farnoud**[*]
Dept. of Electrical & Computer Engineering
University of Virginia
Charlottesville, VA 22903
farzad@virginia.edu

## Abstract

Ranking from noisy comparisons is of great practical interest in machine learning. In this paper, we consider the problem of recovering the exact full ranking for a list of items under ranking models that do *not* assume the Strong Stochastic Transitivity property. We propose a $\delta$-correct algorithm, Probe-Rank, that actively learns the ranking from noisy pairwise comparisons. We prove a sample complexity upper bound for Probe-Rank, which only depends on the preference probabilities between items that are adjacent in the true ranking. This improves upon existing sample complexity results that depend on the preference probabilities for all pairs of items. Probe-Rank thus outperforms existing methods over a large collection of instances that do not satisfy Strong Stochastic Transitivity. Thorough numerical experiments in various settings are conducted, demonstrating that Probe-Rank is significantly more sample-efficient than the state-of-the-art active ranking method.

## 1 Introduction

Ranking from noisy comparisons has a wide range of applications including voting [5, 7], identifying the winner/full ranking of teams in sport leagues, ranking players in online gaming systems [17], crowdsourcing services [6], web search [8], and recommendation systems [2, 23]. In practice, comparisons usually contain certain levels of "noise". For example, duels in a game are not always won by the more proficient player, and preferences between movies/restaurants can also vary among different individuals. The presence of noise is commonly studied using a probabilistic comparison model [12, 25], where an item has a certain probability to win the comparison over another or a group of items.

We are interested in estimating the total ranking. To guarantee that the ranking is consistent with the preference probabilities, it is often assumed [12, 14, 21, 25] that if $i$ ranks higher than $j$, then $i$ wins

---

[*]Co-corresponding Authors

36th Conference on Neural Information Processing Systems (NeurIPS 2022).

a comparison against $j$ with probability $p_{i,j} > \frac{1}{2}$. This assumption is referred to as *Weak Stochastic Transitivity (WST)*. It is clear that the closer $p_{i,j}$ is to $\frac{1}{2}$, the more difficult it becomes to compare $i$ and $j$. A more strict assumption, *Strong Stochastic Transitivity (SST)*, is also often made [10, 24, 26]. SST requires items that have closer ranks to be more difficult to compare, i.e., if $i \succ j \succ k$, then $p_{i,k} \geq \max(p_{i,j}, p_{j,k}) > \frac{1}{2}$. Formal definitions of WST and SST are stated in Section 2.

However, SST can be too strong in many scenarios. For instance, in sports, match outcomes are usually affected by team tactics. Team $k$ may play a tactic that counters team $i$, resulting in a higher winning rate against team $i$ compared with team $j$. Furthermore, items usually have multidimensional features and people may compare different pairs based on different features. A close pair in the overall ranking is thus not necessarily harder to compare than a pair that has a large gap. For example, when comparing cars, people might compare a given pair based on their interior design and another pair based on performance. As another example, in an experiment with games of chance with different probabilities of winning and payoffs [30], it was observed that "people chose between adjacent gambles according to the payoff and between the more extreme gambles according to probability, or expected value."

Motivated by such applications, in this paper, we study the problem of recovering the full ranking of $n$ items under a more general setting, where only WST holds, while SST is not assumed to hold. We focus on only pairwise queries as they are easier to obtain and less prone to error in practice. Furthermore, as many applications [6, 22] allow interactions between users/annotators, we consider comparisons collected in an adaptive manner. Our goal is to use as few comparisons as possible and achieve a high confidence.

Existing algorithms [21, 25] cannot avoid comparing every item $i$ with the item $i^*$ that is the most similar to $i$, i.e., $\left|p_{i,i^*} - \frac{1}{2}\right| = \min_{j \neq i}\{\left|p_{i,j} - \frac{1}{2}\right|\}$. Further, [25] pointed out that comparing item pairs that are adjacent in the true ranking are necessary. When SST holds, adjacent pairs are also the most difficult pairs to distinguish, existing methods thus achieve sample-efficiency. For example, the Iterative-Insertion-Ranking (IIR) algorithm proposed in [25] maintains a preference tree and performs ranking by inserting items one after another. During the insertion process, every item is possible to be compared with every other item (and thus the most similar one), depending on the relative order of insertion and the true ranking. Under SST, IIR was shown to enjoy the optimal sample complexity with mild conditions.

However, when SST does not hold, comparing nonadjacent items harms the performance. Consider an extreme scenario where the true ranking is $1 \succ 2 \succ 3$ and $p_{1,2} = p_{2,3} = 0.8, p_{1,3} = \frac{1}{2} + 2^{-10}$. If item 1 is directly compared to item 3, then it can take $\Theta\left(2^{20}\right)$ queries[2]. For instance, in IIR, this can happen during the insertion process of item 3 when item 1 happens to be the root of the preference tree. A simple fix exists as we can let the three pairs be compared simultaneously. The comparisons between items 1 and 2, items 2 and 3 will terminate much earlier and provide us with the accurate enough information $1 \succ 2, 2 \succ 3$, which is sufficient to recover the total ranking. Therefore, it is important to devise an algorithm whose sample complexity will not be harmed when SST fails to hold.

**Contribution.** In this paper, we propose an active algorithm, termed *Probe-Rank*, that ranks $n$ items based on pairwise comparisons. Probe-Rank is a maxing-based algorithm, i.e., it ranks items by performing $n-1$ steps of maxing. We show that as long as the WST condition is satisfied, with probability at least $1 - \delta$, Probe-Rank returns the correct ranking after conducting at most

$$O\left(n \sum_{i=1}^{n} \left(\widetilde{\Delta}_i^{-2}\right) \left(\log\log\left(\widetilde{\Delta}_i^{-1}\right) + \log(n/\delta)\right)\right) \tag{1}$$

comparisons, where $\widetilde{\Delta}_i = \min_{j:j \text{ and } i \text{ are adjacent}} \left|p_{i,j} - \frac{1}{2}\right|$. Probe-Rank is the first algorithm whose sample complexity only depends on comparison probabilities of adjacent items instead of all pairs of items [21, 25, 29, 31]. Theoretical analyses and numerical experiments under various settings are provided and show that Probe-Rank is more efficient than the state-of-the-art methods when comparing nonadjacent items is more difficult than comparing adjacent items. We also present a preliminary analysis on the sample complexity lower bound in the worst case scenario when SST

---

[2]In fact, according to [13], we need $\Theta\left((p_{i,j} - 1/2)^{-2}\right)$ comparisons to be confident enough about the order between any two items $i$ and $j$ , $i, j \in [n]$.

does not hold. Further, we present a variant of Probe-Rank, named Probe-Rank-SE, in Appendix B. Numerical experiments show that the variant is more sample-efficient under various settings.

## 2 Preliminaries

**Notation**  Without loss of generality, let $[n] = \{1, 2, \ldots, n\}$ denote the set of $n$ items. We write $p \sim Uni(a, b)$ to denote that $p$ is sampled uniformly at random from the interval $(a, b)$, and use $\text{Ber}(p)$ to denote a Bernoulli random variable which equals 1 with probability $p$. We use $(i, j)$ to denote the unordered item pair, i.e., $(i, j) = (j, i)$. Comparisons between items are probabilistic. Whenever two items $i, j$ are compared, $i$ (respectively, $j$) is preferred with probability $p_{i,j}$ (respectively, $p_{j,i}$), independent of any other quantities. For all $i \neq j$, $p_{i,j} + p_{j,i} = 1$. A probabilistic comparison model over $[n]$ is thus defined by the set of probabilities $\mathbf{P} = \{p_{i,j}\}_{1 \leq i < j \leq n}$.

In this paper, asymptotic notation including $O(\cdot), \Omega(\cdot), \Theta(\cdot)$ are defined in the standard sense, with $\widetilde{O}(\cdot), \widetilde{\Omega}(\cdot), \widetilde{\Theta}(\cdot)$ denoting corresponding weaker forms by allowing logarithmic factors.

**Problem setup**  We assume that there exists a total ordering '$\succ$' over $[n]$ such that $\sigma_1 \succ \sigma_2 \succ \cdots \succ \sigma_n$ for some permutation $\sigma = (\sigma_1, \ldots, \sigma_n)$ of $[n]$. The permutation $\sigma$ is referred to as the true ranking. Two items are called adjacent if they are adjacent in $\sigma$, i.e., one ranks right next to the other. To ensure that the true ranking $\sigma$ is consistent with comparisons, we also assume that $i$ has a higher rank than $j$ if and only if $p_{i,j} > \frac{1}{2}$. In other words, if an item $i$ is more preferred than $j$ in $\sigma$, then $i$ has a better chance to win the comparison with $j$. This assumption is known as *Weak Stochastic Transitivity (WST)*. A more strict assumption, *Strong Stochastic Transitivity (SST)*, is also frequently adopted. In addition to WST, SST assumes that whenever $i \succ j \succ k$, $p_{i,k} \geq \max(p_{i,j}, p_{j,k})$. In this paper, we assume only WST and our goal is to recover the true ranking $\sigma$ with a given confidence level $\delta$ by taking pairwise comparisons and minimize the sample complexity. Problem instances are uniquely determined by the permutation $\sigma$ representing the true ranking and the comparison probabilities $\mathbf{P}$.

**Definition 1** ($\delta$-correct algorithm). *An algorithm is said to be $\delta$-correct if for any input instance, with probability at least $1 - \delta$, it returns a correct result in finite time.*

It is clear that the closer $p_{i,j}$ is to $\frac{1}{2}$, the more difficult it becomes to obtain the ordering between $i$ and $j$. Therefore, the probability gap $\Delta_{i,j}$, defined as $\Delta_{i,j} = \left| p_{i,j} - \frac{1}{2} \right|$, provides a characterization of the ranking task difficulty and will be used as a parameter for measuring sample complexities of algorithms. For instance, [25, lemma 12] shows that for any $\delta$-correct algorithm $\mathcal{A}$, $\limsup_{\Delta \to 0} \frac{T_{\mathcal{A}}[\Delta]}{\Delta^{-2}(\log\log \Delta^{-1} + \log \delta^{-1})} > 0$, where $T_{\mathcal{A}}[\Delta]$ is the expected number of samples taken by $\mathcal{A}$ on two items with probability gap $\Delta$. Further, for each item $i$, we define

$$\Delta_i = \min_{j:j\neq i} \Delta_{i,j}, \tag{2}$$

the minimum probability gap between item $i$ and any other item $j$, and define

$$\widetilde{\Delta}_i = \min_{j:j \text{ and } i \text{ are adjacent in } \sigma} \Delta_{i,j}, \tag{3}$$

the minimum probability gap between $i$ and its adjacent items in the true ranking. Note that $\Delta_i \leq \widetilde{\Delta}_i$ by definition and the equality holds when SST is satisfied.

## 3 Related work

The problem of ranking under coherent probabilistic comparisons dates back to 1994 [14]. Feige et al. [14] studied the comparison model assuming that $i \succ j \Leftrightarrow p_{i,j} = \frac{1}{2} + \Delta$ for some known $\Delta$. It was shown that any $\delta$-correct algorithm finds the true ranking with at least $\Theta(n\Delta^{-2}\log(n/\delta))$ comparisons in the worst case. Later in [21], a $\delta$-correct algorithm TOP was proposed to rank the top-$k$ elements by assuming only the existence of a total ranking (WST). The state-of-the-art IIR algorithm was proposed in [25], as discussed in Section 1. A comparison of related algorithms are presented in Table 1.

Table 1: $\delta$-correct algorithms for exact ranking with sample complexity guarantee. Definitions of $\Delta_{i,j}, \Delta_i, \widetilde{\Delta}_i$ can be found in Section 2.

| Algorithm | Assumptions on $\mathbf{P}$ | Sample complexity |
|---|---|---|
| Single Elimination Tournament [21] | WST | $O\left(\frac{n(\log n)^2 \log(1/\delta)}{\min_{1 \le i < j \le n} \Delta_{i,j}^2}\right)$ |
| PLPAC-AMPR [29] | The Plackett-Luce model | $O\left(n \log n \max_{i \in [n]}\left\{\frac{1}{\Delta_i^2} \log(\frac{n}{\delta \Delta_i})\right\}\right)$ |
| Iterative-Insertion-Ranking [25] | WST | $O\left(\sum\limits_{i=1}^{n} \frac{1}{\Delta_i^2}\left(\log\log\frac{1}{\Delta_i} + \log\frac{n}{\delta}\right)\right)$ |
| Probe-Rank (this paper) | WST | $O\left(n \sum\limits_{i=1}^{n} \frac{1}{(\widetilde{\Delta}_i)^2}\left(\log\log\frac{1}{\widetilde{\Delta}_i} + \log\frac{n}{\delta}\right)\right)$ |

Ranking or maxing has also been widely studied under more strict assumptions, e.g., SST, RST[3] and STI[4] and usually in the probably approximately correct (PAC) setting [10, 11, 12, 24, 26, 27, 29, 32]. In particular, [24, 26, 27, 29] considered parametric comparison models such as the multinomial logit (MNL) model. Note that parametric models are often more restrictive and can imply SST/STI conditions. In the PAC setting, the goal is to find an $\epsilon$-ranking $r_1 \succ r_2 \succ \cdots \succ r_n$ such that $p_{r_i, r_j} > \frac{1}{2} - \epsilon$ for all $i < j$. Although $\epsilon$-rankings become closer to the true ranking as $\epsilon$ goes to 0, it is pointed out by [25] that PAC ranking algorithms cannot be easily extended to the case when $\epsilon = 0$. Among all, [12] is the most relevant work to this paper. In [12], PAC ranking and maxing were studied for both SST and WST settings. For WST, an instance-independent lower bound $\Theta(n^2)$ was proved, and a brute-force algorithm which compares each pair to an accuracy of $\epsilon$ and thus conducts $O((n^2/\epsilon^2)\log(n/\delta))$ comparisons was proposed. Note that in this paper, we are aiming at recovering the exact ranking instead of an $\epsilon$-ranking. An exact ranking is preferred over an epsilon-ranking in competitive applications like voting and sport games, where people are not satisfied with an approximate winner. Furthermore, as suggested by [25], analyzing the exact ranking helps us to gain a better understanding about the instance-wise upper and lower bounds. A trivial extension of the brute-force algorithm can lead to sample complexity $\widetilde{O}\left(\frac{n^2}{\min_{i,j} \Delta_{i,j}^2}\right)$, which is substantially worse than our proposed algorithm.

Although we believe WST can be considered a natural and reasonably weak assumption, there are situations that WST does not hold as a ranking over items may not exist or, if it does, all comparison probabilities are not necessarily consistent with that ranking. So another line of research is to allow comparison probabilities $p_{i,j}$ take any values in $(0,1)$ as long as $p_{i,j} + p_{j,i} = 1$. In such scenarios, rankings can be defined and derived based on various criteria including Borda score [16, 19, 28] and Copeland score [4, 33]. The ranking problem has also been studied from a heterogeneous perspective [18, 31], where queries are made by multiple agents with different comparison probabilities. In [15], the problem of testing whether the WST condition holds was studied. More broadly, the problems of ranking, maxing or selection can be formulated in the context of dueling bandits. A comprehensive survey can be found in [3].

## 4 Proposed algorithm

In this section, we propose a $\delta$-correct algorithm for exact ranking of all problem instances that satisfy the WST condition. As mentioned previously, our algorithm is designed to outperform existing methods in situations where nonadjacent items can be more difficult to compare than adjacent items.

To avoid spending unnecessary samples on item pairs with small probability gaps, we propose a subroutine named *Successive-Comparison* (SC) (see Subroutine 1). SC uses a parameter $\tau$ for controlling to what extent the comparison should last. Specifically, SC compares a given item pair for a fixed number $b_\tau = \lceil (2/\epsilon_\tau^2) \log(1/\delta_\tau) \rceil$ times with an accuracy level $\epsilon_\tau = 2^{-\tau}$ and confidence level $\delta_\tau = 6\delta/(\tau^2 \pi^2)$. If the empirical probability that $i$ (respectively, $j$) wins is over $1/2$ by more

---
[3]Under relaxed stochastic transitivity (RST), it is assumed that for all $i \succ j \succ k$, $\Delta_{i,k} \ge \gamma \max\{\Delta_{i,j}, \Delta_{j,k}\}$ for some $0 < \gamma < 1$.

[4]Under stochastic triangle inequality (STI), it is assumed that for all $i \succ j \succ k$, $\Delta_{i,k} \le \Delta_{i,j} + \Delta_{j,k}$.

than $\epsilon_\tau/2$, then SC returns $i$ (respectively, $j$) as the more preferred item. Otherwise, SC will return 'unsure' to inform us that more samples are needed.

For two items $i$ and $j$, SC $(i, j, \delta, \tau)$ will be called successively with $\tau$ increasing by 1 at a time. We show in Appendix A that after $\tau$ gets large enough such that $\epsilon_\tau \leq \Delta_{i,j}$, the correct ordering between $i$ and $j$ will be returned with high probability.

---

**Subroutine 1** Successive-Comparison$(i, j, \delta, \tau)$ (SC)

1: **Input:** items $i, j$, confidence level $\delta$, probing parameter $\tau$
2: $w_i = 0, \epsilon_\tau = 2^{-\tau}, \delta_\tau = \frac{\delta}{c\tau^2}, c = \frac{\pi^2}{6}, b_\tau = \left\lceil \frac{2}{\epsilon_\tau^2} \log \frac{1}{\delta_\tau} \right\rceil$;
3: **For** $t = 1$ to $b_\tau$ **do**
4:     compare $i$ and $j$ once; if $i$ wins, $w_i = w_i + 1$;
5: $\widehat{p}_i = w_i/b_\tau$;
6: **return** $[i, j]$ **if** $\widehat{p}_i - \frac{1}{2} > \frac{1}{2}\epsilon_\tau$; **return** $[j, i]$ **if** $\widehat{p}_i - \frac{1}{2} < -\frac{1}{2}\epsilon_\tau$; and **return** 'unsure' **else**;

---

**Partial order preserving graph** During the ranking process, we maintain a directed graph $T$ to store the partial orders we have obtained from SC instances so far. The graph $T$ is initialized with $n$ nodes $V_1, \ldots, V_n$ and no edge exists between any two nodes. Nodes $V_1, V_2, \ldots, V_n$ represent items $1, 2, \ldots, n$, respectively. In our algorithm, $T$ is involved with three types of operations, *edge update*, *node removal* and *maximal set selection*. Every time an instance of SC returns a pairwise order, e.g., $i \succ j$, we add a directed edge from $V_i$ to $V_j$, written as $T = T \cup (i \succ j)$. Moreover, we also complete all edges in the transitive closure of the existing edges. In other words, if the edge between $V_i$ and $V_j$ induces a directed path from $V_{k_1}$ to $V_{k_2}$, then a directed edge from $V_{k_1}$ to $V_{k_2}$ is also added to $T$. By completing the transitive closure, we can avoid comparing pairs whose ordering can be inferred from current knowledge and keep $T$ acyclic. In the ranking process, we only run comparisons on item pairs that are not connected by edges and hence no contradictions in orderings will be returned by SC. By removing node $V_i$, we remove $V_i$ and all edges of $V_i$ from $T$. The maximal elements of $T$ are the nodes which do not have any incoming edges. Since edges represent comparison results returned by SC, maximal elements correspond to items that have not lost to any other items. Note that since $T$ is acyclic, maximal elements always exist.

Next, we establish our ranking algorithm *Probe-Rank* (see Algorithm 2). Probe-Rank finds the true ranking by performing maxing for $n - 1$ rounds. In every round $t$, subroutine *Probe-Max* returns an item in $S_t$ as the most preferred item (the maximum), where $S_t$ denotes the set of remaining unranked items right before round $t$. The strategy of Probe-Max is to repeatedly apply SC on all item pairs. For every item pair $(i, j)$, we initialize a global variable $\tau_{i,j}$ as the probing parameter for SC instances that run over $i, j$. The graph $T$ storing obtained partial orders is also viewed as a global variable. Parameters $\tau_{i,j}$ and graph $T$ will be accessed and altered in Probe-Max.

---

**Algorithm 2** Probe-Rank

1: **Input:** items $[n]$, confidence level $\delta$
2: $S_1 = [n], Ans = [0]^n$, initialize $T, \tau_{i,j} = 1$ for all pairs of items $i \neq j$;
3: **For** $t = 1$ to $n - 1$ **do**
4:     $i_{max} = $ Probe-Max$(S_t, 2\delta/n^2)$;
5:     remove $i_{max}$ from $T$; $Ans[t - 1] = i_{max}$; $S_{t+1} = S_t \setminus \{i_{max}\}$;
6: $Ans[n - 1] = S_n[0]$; **return** $Ans$;

---

In Probe-Max$(S, \delta)$ (see Subroutine 3), SC instances are performed only on items that are possible to be the actual maximum. Let $U$ be the set of maximal elements in $T$. By definition, every item in $U$ has not lost to any other item in $S$ yet. Assuming all previous comparison results (obtained form SC) are correct, to find the actual maximum, it suffices to focus on items in $U$. We use $S^2$ to denote the set of all unordered item pairs in $S$, i.e., $S^2 = \{(a, b) : a, b \in S, a \neq b\}$. All 'legitimate' pairs that can potentially provide us with information about the maximum item in $S$ are thus

$$P = \{(i, j) : (i \in U \text{ or } j \in U), (i, j) \in S^2, (i, j) \notin T\}, \tag{4}$$

where $(i, j) \notin T$ means that nodes $V_i$ and $V_j$ are not connected in $T$. While $U$ contains more than one items, Probe-Max keeps applying SC on item pairs in $P$. If an item in $U$ loses a comparison, then

we remove it from $U$. In every iteration of the while loop, the pairs $(i^*, j^*)$ in $P$ with the smallest $\tau$ value are chosen and SC $(i^*, j^*, \delta, \tau_{i^*, j^*})$ are performed. Note that the $\tau$ value increases by one after each call of SC. Starting with item pairs with small $\tau$ values guarantees that we do not miss any useful information that can be obtained by paying only a small amount of comparisons.

---

**Subroutine 3** Probe-Max$(S, \delta)$

---

1: **Input:** set of unranked items $S$, SC confidence level $\delta$
2: Let $U$ be the set of maximal elements according to $T$;
3: **While** $|U| > 1$ **do**
4:     Let $P = \{(i, j) : (i \in U \text{ or } j \in U), (i, j) \in S^2, (i, j) \notin T\}$;
5:     **For** $(a, b)$ in $\mathrm{argmin}_{(x, y) \in P} \tau_{x, y}$ **do**
6:         $Ans = \mathrm{SC}(a, b, \delta, \tau_{a, b})$; $\tau_{a, b} = \tau_{a, b} + 1$;
7:         **If** $Ans$ is not 'unsure' **then**
8:             $w, l = Ans$; $T = T \cup (w \succ l)$; **If** $|U| > 1$ and $l \in U$ **then** $U = U \setminus \{l\}$;
9: **return** $U[0]$;

---

We provide a simple example demonstrating the ranking process.

**Example 1.** Consider items $\{1, 2, 3, 4\}$ with true ranking $1 \succ 2 \succ 3 \succ 4$. Figure 1 shows the status of $T, U, S_t$ throughout the ranking process. In particular, we assume the pairwise comparison results are all correct and returned in order $1 \succ 2, 2 \succ 4, 1 \succ 3, 2 \succ 3, 3 \succ 4$.

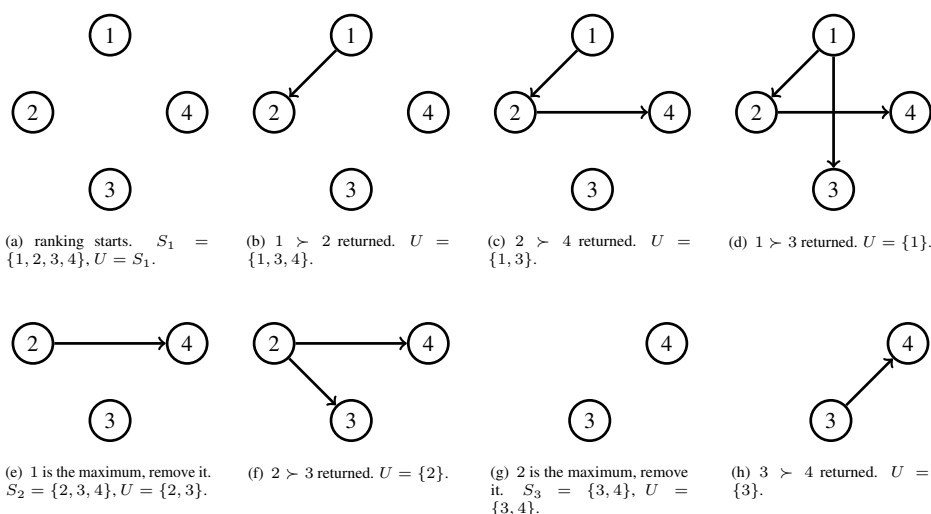

(a) ranking starts. $S_1 = \{1, 2, 3, 4\}, U = S_1$.

(b) $1 \succ 2$ returned. $U = \{1, 3, 4\}$.

(c) $2 \succ 4$ returned. $U = \{1, 3\}$.

(d) $1 \succ 3$ returned. $U = \{1\}$.

(e) 1 is the maximum, remove it. $S_2 = \{2, 3, 4\}, U = \{2, 3\}$.

(f) $2 \succ 3$ returned. $U = \{2\}$.

(g) 2 is the maximum, remove it. $S_3 = \{3, 4\}, U = \{3, 4\}$.

(h) $3 \succ 4$ returned. $U = \{3\}$.

Figure 1: An illustration of the steps by Probe-Ranking, assuming true ranking as $1 \succ 2 \succ 3 \succ 4$.

## 5 Upper bound on the sample complexity of Probe-Rank

In this section, we provide a sample complexity upper bound for the proposed algorithm Probe-Rank.

**Theorem 2.** *Let $\delta > 0$ be an arbitrary constant. For all problem instances satisfying the Weak Stochastic Transitivity (WST) property, with probability at least $1 - \delta$, Probe-Rank returns the true ranking of $n$ items and conducts at most*

$$O\left(n \sum_{i=1}^{n} \left(\widetilde{\Delta}_i^{-2}\right) \left(\log\log\left(\widetilde{\Delta}_i^{-1}\right) + \log\left(\frac{n}{\delta}\right)\right)\right) \tag{5}$$

*comparisons, where $\widetilde{\Delta}_i$ is defined as in* (3).

The proof of Theorem 2 is deferred to Appendix A.

By the preceding theorem, the sample complexity of Probe-Rank is upper bounded by the sum of terms $(\widetilde{\Delta}_i)^{-2}(\log\log(\widetilde{\Delta}_i)^{-1} + \log(n/\delta))$ with an additional multiplicative factor of $n$. Recall from Section 2 that the term $(\widetilde{\Delta}_i)^{-2}(\log\log(\widetilde{\Delta}_i)^{-1} + \log(n/\delta))$ can be viewed as a lower bound on the number of comparisons that is needed for obtaining the order between $i$ and its adjacent items with confidence level $\delta/n$. Theorem 2 thus suggests that in Probe-Rank, every item is compared until it can be distinguished from its neighbors and no further. This matches with our intuition that only comparisons between adjacent items are necessary, and a single nonadjacent pair being extremely hard to distinguish should not harm the overall sample complexity. In contrast, sample complexities of existing algorithms are determined by the smallest probability gap between items, which can lead to a substantially large amount of unnecessary comparisons.

However, Probe-Rank achieves the dependence on $\widetilde{\Delta}_i$ instead of $\Delta_i$ at the cost of an additional multiplicative factor of $n$. Intuitively, because we have zero prior information about which items are adjacent and which are not, Probe-Rank pays $\Theta(n)$ attempts for each item $i$ in order to 'identify' its neighbors and get the ordering feedback.

We compare Probe-Rank with the state-of-the-art IIR algorithm. Let $\mathcal{C}$ (Probe) and $\mathcal{C}$ (IIR) denote the sample complexities of two algorithms. From Table 1 and Theorem 2,

$$\mathcal{C}\,(\text{Probe}) = \sum_{i=1}^{n} \widetilde{\Theta}\left(n(\widetilde{\Delta}_i)^{-2}\right), \quad \mathcal{C}\,(\text{IIR}) = \sum_{i=1}^{n} \widetilde{\Theta}\left((\Delta_i)^{-2}\right), \qquad (6)$$

noting that from the proofs, the sample complexity upper bounds are both tight in the worst case.

Under WST with no other conditions assumed, $\Delta_i \le \widetilde{\Delta}_i$. In particular, when $\widetilde{\Delta}_i/\Delta_i = \Theta(\sqrt{n})$ for all $i$, then $\mathcal{C}$ (Probe) and $\mathcal{C}$ (IIR) are of the same asymptotic order with respect to $n$; if $\widetilde{\Delta}_i/\Delta_i = \omega(\sqrt{n})$, then Probe-Rank is asymptotically more sample-efficient than IIR. These phenomena are also reflected in our numerical experiments in Section 6 (see Figure 3).

**Remark.** It is worth noting that IIR is optimal if the more strict assumption SST as well as some other conditions are made, as shown in [25]. When SST holds, $\widetilde{\Delta}_i = \Delta_i$. Probe-Rank thus suffers from an additional factor of $n$. This case is also included in our numerical experiment (see Figure 2(a)).

## 6 Experiments

In this section, we present numerical experiments demonstrating the practical performance of Probe-Rank. We compare Probe-Rank with the IIR algorithm, which was shown to outperform all the other baseline algorithms both theoretically and numerically [25]. Our implementation can be found on Github [5].

We study different settings where SST is satisfied, not guaranteed, or violated, but WST always holds, which is consistent with our theory. Specifically, we want to rank $n$ items with the true ranking $\sigma_1 \succ \sigma_2 \succ \cdots \succ \sigma_n$, where $n$ varies over $[10, 100]$. The probabilistic comparison model $p_{ij}$ is generated in different ways to satisfy different assumptions. Note that $\Delta$ and $\Delta_d$ are tuning parameters in all the following settings.

- `SST`: SST is satisfied. Comparison probabilities $p_{ij}$ are generated from the MNL model, where $p_{\sigma_i,\sigma_j} = (\exp(s_{\sigma_i} - s_{\sigma_j}) + 1)^{-1}$, and $s_{\sigma_1}, \ldots, s_{\sigma_n}$ is a decreasing sequence where $s_{\sigma_i} = 100\Delta_d \cdot \frac{(n+1-i)}{n}$.
- `WST`: SST does not necessarily hold. Let $p_{i,j} \sim Uni(\frac{1}{2} + \Delta_d, 1)$ for all items $i \succ j$.
- `NON-SST`: SST does not hold. For adjacent items, we have $p_{\sigma_i,\sigma_{i+1}} \sim Uni\left(\frac{1}{2} + \Delta_d, 1\right)$. Otherwise, we have $p_{\sigma_i,\sigma_j} \sim Uni\left(\frac{1}{2} + \frac{\Delta_d}{10}, \frac{1}{2} + \Delta_d\right)$ for $j > i + 1$.
- `ADJ-ASYM`: SST does not hold. This setting is used to verify the asymptotic analysis in Section 5. For adjacent items, we set $p_{\sigma_i,\sigma_{i+1}} = \frac{1}{2} + \Delta_d$. Otherwise, we set $p_{\sigma_i,\sigma_j} = \frac{1}{2} + \frac{\Delta_d}{n^\alpha}$ for $j > i + 1$. We consider cases where $\alpha$ equals 0.5 or 1.
- `ADJ-CNST`: SST does not hold. For adjacent items, we set $p_{\sigma_i,\sigma_{i+1}} = \frac{1}{2} + \Delta$. Otherwise $p_{\sigma_i,\sigma_j} = \frac{1}{2} + \Delta_d$ for $j > i + 1$. Here $\Delta > \Delta_d$.

[5] https://github.com/tao-j/aht/releases/tag/v0.1

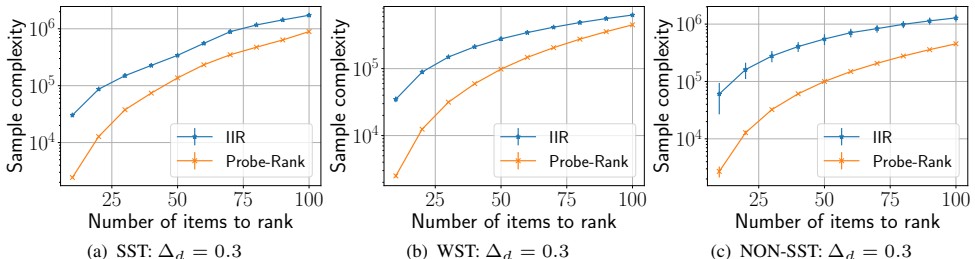

(a) SST: $\Delta_d = 0.3$     (b) WST: $\Delta_d = 0.3$     (c) NON-SST: $\Delta_d = 0.3$

Figure 2: Comparison of sample complexities of Probe-Rank and IIR under various settings. In each subfigure, $\Delta_d$ is fixed while the number of items varies.

All experiments are averaged over 100 independent trials. For each trial, the ground truth ranking $\sigma$ is generated uniformly at random and the comparison probabilities are assigned accordingly. The confidence level $\delta$ is fixed to be 0.1. Throughout the experiment, every trial for every algorithm successfully recovered the correct ranking.

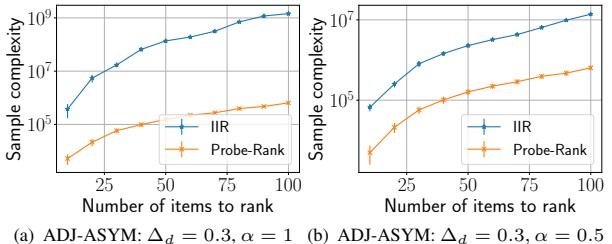

(a) ADJ-ASYM: $\Delta_d = 0.3, \alpha = 1$    (b) ADJ-ASYM: $\Delta_d = 0.3, \alpha = 0.5$

Figure 3: Relationship between $n$ and gap $\Delta_d$

We use internal clusters of intel "Skylake" generation CPUs. Each job contains a single model type for item numbers ranging from 10 to 100 with a step size of 10. Models are generated from a job unique random seed shared among the two algorithms. Most jobs with sample complexity smaller than $10^7$ terminate in 3 minutes. For $\Delta_d = 0.1$ under the `ADJ-ASYM` model, 3 hours are needed due to high sample complexity. Due to the space limit, more detailed experimental setups and thorough ablation studies can be found in Appendix C.

**Performance comparison** Figure 2 with y-axis in log-scale shows comparison of IIR and Probe-Ranking under the `SST`, `WST` and `NON-SST` settings. The parameter $\Delta_d$ is set to be 0.3. It can be seen that under the `SST` and `WST` settings (Figures 2(a), 2(b)), Probe-Rank consumes less samples than IIR for small $n$. As $n$ gets larger, however, IIR becomes more sample-efficient due to that Probe-Rank has an additional factor of $n$ in its sample complexity compared with IIR for instances satisfy SST. However, under the `NON-SST` setting where SST does not hold, Probe-Rank has a clear advantage over IIR, as shown in Figure 2(c).

**Dependence on $n$ and the probability gaps** Following Theorem 2, we verify that the sample complexity of Probe-Rank is lower than IIR when the number of items $n$ gets larger. We use the

`ADJ-ASYM` setting to simulate situations where nonadjacent items can be much more difficult to compare. In particular, we choose $\alpha = 1$ (see Figure 3(a)) and $\alpha = 1/2$ (see Figure 3(b)). It can be seen from Figure 3(a) that as the number of items $n$ gets larger, the gap between the two curves also gets larger. This matches our analysis that when $\widetilde{\Delta}_i/\Delta_i = \omega(\sqrt{n})$, then the sample complexity of IIR is of higher order than that of Probe-Rank. When $\widetilde{\Delta}_i/\Delta_i = \Theta(\sqrt{n})$, Figure 3(b) shows that the gap between the two sample complexities varies little as $n$ increases.

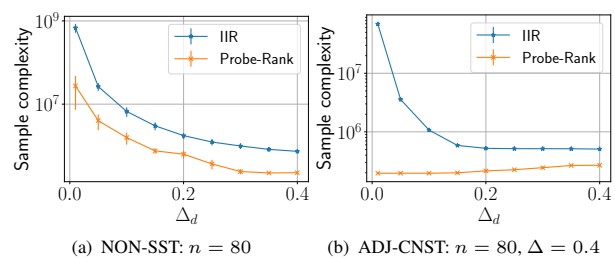

(a) NON-SST: $n = 80$    (b) ADJ-CNST: $n = 80, \Delta = 0.4$

Figure 4: Ablation study on the dependence of the sample complexity on the probability gap $\Delta_d$.

Our analysis also suggests that sample complexities of two algorithms are of the same order.

Furthermore, we show through the `NON-SST` and `ADJ-CNST` settings that when the probability gaps of nonadjacent item pairs decrease, the advantage of our algorithm will be more and more prominent.

In Figure 4, we fix $n = 80$ and let $\Delta_d$ vary. Clearly, Probe-Rank has an advantage over IIR in both settings. In particular, Figure 4(b) shows the comparison of two algorithms in the `ADJ-CNST` setting with the probability gaps between adjacent items $\Delta$ fixed as $0.4$. As the probability gap between nonadjacent items $\Delta_d$ varies from $0.01$ to $0.4$, it can be seen that the sample complexity of Probe-Rank does not vary much. However, the sample complexity of IIR has a positive correlation with $\frac{1}{\Delta_d^2}$. This numerical result matches our analysis that Probe-Ranking is not affected by the comparison probability of nonadjacent items, which does not hold for IIR.

## 7 Discussion on the lower bound

In this section, we provide some insights about the lower bound for pairwise ranking by proposing a conjecture based on a particularly hard instance $\mathcal{I}_{WST}$ that satisfies the WST condition.

**Problem 1** ($\mathcal{I}_{WST}$). The problem instance $\mathcal{I}_{WST}$ is constructed as follows. Consider $n$ items with an underlying ordering '$\succ$'. For all $i \succ j$,

$$
p_{i,j} = \begin{cases} \frac{1}{2} + \Delta, & \text{if } i \text{ and } j \text{ are adjacent,} \\ \frac{1}{2} + cn^{-10}\Delta^2/\log(1/\delta), & \text{otherwise,} \end{cases}
$$

where $c$ and $\Delta$ are constants and $n^{-10}$ can be replaced by any other quantity that is smaller than $n^{-2}$.

By a reduction, any $\delta$-correct algorithm that finds the maximum item for $\mathcal{I}_{WST}$ can be constructed to find the maximum item for $\mathcal{I}_{SNG}$, described below in Problem 2. Therefore, a lower bound on the sample complexity for maxing in Problem 2 will imply a lower bound of the same order for the maxing (and thus, ranking) problem for $\mathcal{I}_{WST}$. This lower bound is also a worst-case lower bound for ranking under WST. In the following, we provide an analysis for Problem 2. The reduction technique will be deferred to Appendix D.

**Problem 2** ($\mathcal{I}_{SNG}$). Consider $n$ items with an underlying ordering '$\succ$'. One can make queries of the form 'if $i \succ j$'. The feedback $Y_{i,j}$ is a binary random variable which takes value 1 if the answer is YES and takes value 0 otherwise. The random variables $Y_{i,j}$ are defined to follow distributions:

$$
Y_{i,j} \sim \begin{cases} \text{Ber}(\frac{1}{2} - 2\Delta), & \text{if } i \prec j \text{ and } i, j \text{ are adjacent,} \\ \text{Ber}(\frac{1}{2}), & \text{otherwise.} \end{cases}
$$

Consider random vectors defined by $\boldsymbol{p}_i = (Y_{i,1}, Y_{i,2}, \ldots, Y_{i,n})$ in Problem 2. The maximum element $i^*$ corresponds to the random vector $\boldsymbol{p}_{i^*}$, where each entry is a $1/2$-Bernoulli random variable. For every other non-maximum element $i$, $\boldsymbol{p}_i$ will contain exactly one $(1/2 - 2\Delta)$-Bernoulli random variable. Under such problem setting, finding the maximum item is equivalent to finding which vector has all its entries as $1/2$-Bernoulli random variables.

We conjecture that any $\delta$-correct algorithm that can find the maximum item for $\mathcal{I}_{SNG}$ has a sample complexity at least

$$
\Omega\big(n^2\Delta^{-2}\log(1/\delta)\big). \tag{7}
$$

We start from viewing it as a hypothesis testing problem. Consider that an agent is asked to determine if $\boldsymbol{p}_1$ satisfies hypothesis $H_0$, defined as

$$
H_0 : \boldsymbol{p}_1 = (p_{1,1}, \ldots, p_{1,n}), \text{ where } p_{1,k} \sim \text{Ber}(1/2), \forall k \in [n],
$$

or $H_j$, in which the $j$-th entry is biased:

$$
H_j : \boldsymbol{p}_1 = (p_{1,1}, \ldots, p_{1,n}), \text{ where } p_{1,j} \sim \text{Ber}(1/2 - 2\Delta), p_{1,k} \sim \text{Ber}(1/2), \forall k \neq j.
$$

Suppose the hypothesis testing algorithm $\mathcal{A}$ is $\delta$-correct and stops within $T$ rounds of interactions. We denote $\mathcal{A}(T)$ as the output at the $T$-th round, which is either 0 (accept $H_0$) or 1 (reject $H_0$). For any given $j \neq 1$, by the Bretagnolle–Huber inequality, we have

$$
2\delta \geq \mathbb{P}_0(\mathcal{A}(T) \neq 0) + \mathbb{P}_j(\mathcal{A}(T) = 0) \geq \frac{1}{2}e^{-\text{KL}(\mathbb{P}_0^{\mathcal{A}}\|\mathbb{P}_j^{\mathcal{A}})}, \tag{8}
$$

where $\mathbb{P}_0$ is the probability measure under $H_0$, and $\mathbb{P}_0^{\mathcal{A}}$ is the probability measure of the canonical bandit model under $H_0$. In fact, we have the divergence decomposition lemma [20, Lemma 15.1]:

$$\mathrm{KL}(\mathbb{P}_0^{\mathcal{A}}||\mathbb{P}_j^{\mathcal{A}}) = \sum_{k=1}^{n} \mathbb{E}_0[N_k(T)]\mathrm{KL}(\mathbb{P}_{0,k}||\mathbb{P}_{j,k}) = \mathbb{E}_0[N_j(T)]\mathrm{KL}(\mathrm{Ber}(1/2)||\mathrm{Ber}(1/2-2\Delta)), \quad (9)$$

where $\mathbb{E}_0$ denotes the expectation under $H_0$; $\mathbb{E}_0[N_k(T)]$ denotes under $H_0$, the expected number of queries for the entry $p_{1,k}$ within $T$ rounds.; $\mathbb{P}_{0,k}$, $\mathbb{P}_{j,k}$ are the Bernoulli distributions specified by $p_{1,k}$ under $H_0$, $H_j$, respectively. The second equality is due to the fact that the only difference between $H_0$ and $H_j$ is that the $j$-th entry has different Bernoulli distributions.

Combining the two inequality above gives:

$$\mathbb{E}_0[N_j(T)]\mathrm{KL}(\mathrm{Ber}(1/2)||\mathrm{Ber}(1/2-2\Delta)) \geq \log(1/4\delta). \quad (10)$$

Since $\mathrm{KL}(\mathrm{Ber}(1/2)||\mathrm{Ber}(1/2-x)) < (4x)^2$ for all $x < 2/9$, we get $\mathbb{E}_0[N_j(T)] = \Omega(\Delta^{-2}\log(1/\delta))$. Thus, the total expected number of queries under $H_0$ will be $\Omega(n\Delta^{-2}\log(1/\delta))$.

In Problem 2, there are in total $n$ vectors. We reasonably conjecture that to identify which vector satisfies $H_0$ requires at least $\Omega(n)$ attempts, with each attempt costs $\Omega(n\Delta^{-2}\log(1/\delta))$, i.e, any $\delta$-correct algorithm requires $\Omega\left(n^2\log(1/\delta)/\Delta^2\right)$ queries.

## 8   Conclusion and future work

In this paper, we studied the problem of exact ranking under the most general assumption WST. We proposed a $\delta$-correct algorithm Probe-Rank, and derived an instance-wise upper bound on its sample complexity. The upper bound shows that the performance of Probe-Rank only depend on the comparison probabilities of adjacent items and thus improves existing results when SST does not hold. Numerical results also suggest that our ranking algorithm outperforms state-of-the-art. A discussion over the lower bound for pairwise ranking is also provided. We propose a conjecture that in the worst case, any algorithm has sample complexity $n$ times the number of comparisons needed for comparing all adjacent items. However, it remains an open problem whether our conjecture holds and will be left to future work.

## Acknowledgments and Disclosure of Funding

We would like to thank the anonymous reviewers for their helpful comments. HL, TJ and FF are supported in part by the NSF grant CIF-1908544. YW and QG are supported in part by the NSF grant CIF-1911168. The views and conclusions contained in this paper are those of the authors and should not be interpreted as representing any funding agencies.

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
