Since $\text{KL}(\text{Ber}(1/2)||\text{Ber}(1/2-x)) < (4x)^2$ for all $x < 2/9$, we get $\mathbb{E}_0[N_j(T)] = \Omega(\Delta^{-2}\log(1/\delta))$. Thus, the total expected number of queries under $H_0$ will be $\Omega(n\Delta^{-2}\log(1/\delta))$.

In Problem 2, there are in total $n$ vectors. We reasonably conjecture that to identify which vector satisfies $H_0$ requires at least $\Omega(n)$ attempts, with each attempt costs $\Omega(n\Delta^{-2}\log(1/\delta))$, i.e, any $\delta$-correct algorithm requires $\Omega(n^2\log(1/\delta)/\Delta^2)$ queries.

## 8 Conclusion and future work

In this paper, we studied the problem of exact ranking under the most general assumption WST. We proposed a $\delta$-correct algorithm Probe-Rank, and derived an instance-wise upper bound on its sample complexity. The upper bound shows that the performance of Probe-Rank only depend on the comparison probabilities of adjacent items and thus improves existing results when SST does not hold. Numerical results also suggest that our ranking algorithm outperforms state-of-the-art. A discussion over the lower bound for pairwise ranking is also provided. We propose a conjecture that in the worst case, any algorithm has sample complexity $n$ times the number of comparisons needed for comparing all adjacent items. However, it remains an open problem whether our conjecture holds and will be left to future work.

## Acknowledgments and Disclosure of Funding

We would like to thank the anonymous reviewers for their helpful comments. HL, TJ and FF are supported in part by the NSF grant CIF-1908544. YW and QG are supported in part by the NSF grant CIF-1911168. The views and conclusions contained in this paper are those of the authors and should not be interpreted as representing any funding agencies.

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

# A   Proof of the sample complexity upper bound on Probe-Rank

In this section, we prove our theoretical upper bound presented in Theorem 2, Section 5.

We first show in the following lemma that the subroutine Successive-Comparison returns desired outcomes with high probability. Given an item pair $(i, j)$ with probability gap $\Delta_{i,j} > 0$ and a positive integer $\tau$, we say SC $(i, j, \delta, \tau)$ is successful if one of the following two events holds,

$$\mathcal{E}_1 = \{\Delta_{i,j} \geq \epsilon_\tau \text{ and SC correctly returns } [i, j]\}, \tag{11}$$
$$\mathcal{E}_2 = \{\Delta_{i,j} < \epsilon_\tau \text{ and SC returns 'unsure' or } [i, j]\}. \tag{12}$$

**Lemma 3.** *For an item pair $(i, j)$ with probability gap $\Delta_{i,j} > 0$ and a positive integer $\tau$, SC $(i, j, \delta, \tau)$ is successful with probability at least $1 - \frac{\delta}{c\tau^2}$, where $c = \frac{\pi^2}{6}$.*

*Proof of Lemma 3.*   Hoeffding's inequality gives that

$$\Pr\left(\widehat{p}_i - p_{i,j} \leq -\frac{1}{2}\epsilon_\tau\right) \leq \exp\left(-2b_\tau\left(\frac{1}{2}\epsilon_\tau\right)^2\right) \leq \frac{\delta}{c\tau^2}. \tag{13}$$

Therefore, the probability that SC outputs $[j, i]$ is at most

$$\Pr\left(\widehat{p}_i - \frac{1}{2} < -\frac{1}{2}\epsilon_\tau\right) \leq \Pr\left(\widehat{p}_i - p_{i,j} \leq -\frac{1}{2}\epsilon_\tau\right) \leq \frac{\delta}{c\tau^2}, \tag{14}$$

and the probability that SC returns $[i, j]$ or 'unsure' is at least $1 - \frac{\delta}{c\tau^2}$.

Further, if $\Delta_{i,j} \geq \epsilon_\tau$, the probability that SC returns $[i, j]$ is at least

$$\Pr\left(\widehat{p}_i - \frac{1}{2} > \frac{1}{2}\epsilon_\tau\right) = \Pr\left(\widehat{p}_i > \frac{1}{2} + \frac{1}{2}\epsilon_\tau\right) \geq \Pr\left(\widehat{p}_i > p_{i,j} - \frac{1}{2}\epsilon_\tau\right) \geq 1 - \frac{\delta}{c\tau^2}. \tag{15}$$

This completes the proof.                                                                   $\square$

By Lemma 3, with high probability, SC does not return the incorrect ordering. Further, if $\tau$ is large enough, then SC is guaranteed to return the correct ordering. We use Lemma 3 to show the theoretical performance of Probe-Rank.

**Theorem 2.** *Let $\delta > 0$ be an arbitrary constant. For all problem instances satisfying the Weak Stochastic Transitivity (WST) property, with probability at least $1 - \delta$, Probe-Rank returns the true ranking of $n$ items and conducts at most*

$$O\left(n\sum_{i=1}^{n}\left(\widetilde{\Delta}_i^{-2}\right)\left(\log\log\left(\widetilde{\Delta}_i^{-1}\right) + \log\left(\frac{n}{\delta}\right)\right)\right) \tag{5}$$

*comparisons, where $\widetilde{\Delta}_i$ is defined as in (3).*

*Proof of Theorem 2.*   Define events

$$\mathcal{E}_{i,j}(\tau) = \{\text{SC}\left(i, j, 2\delta/n^2, \tau\right) \text{ is successful}\}. \tag{16}$$

Define the bad event

$$\mathcal{E}^{bad} = \cup_{(i,j)\in[n]^2} \cup_{\tau=1}^{\infty} \left(\mathcal{E}_{i,j}(\tau)\right)^c. \tag{17}$$

By the union bound and Lemma 3

$$\Pr\left(\mathcal{E}^{bad}\right) \leq \sum_{(i,j)\in[n]^2} \sum_{\tau=1}^{\infty} \frac{2\delta}{cn^2\tau^2} \leq \sum_{\tau=1}^{\infty} \frac{\delta}{c\tau^2} \leq \delta. \tag{18}$$

In the following, we assume that $\mathcal{E}^{bad}$ does not happen.

**Correctness.** We show that when $\mathcal{E}^{bad}$ does not happen, in every round $t$, Probe-Max$(S_t, 2\delta/n^2)$ (line 4 of Algorithm 2) correctly returns the most preferred item in the set of remaining items $S_t$. Since the probability of $\mathcal{E}^{bad}$ is upper bounded by $\delta$, the correctness of Probe-Rank thus follows.

Let $x$ be the most preferred item in $S_t$. When $\mathcal{E}^{bad}$ does not happen, all comparison results returned by SC are correct and $T$ is always consistent with the true ranking. Thus, no item in $S_t$ is known to rank higher than $x$, i.e., at the beginning of Subroutine 3, $x \in U$. Moreover, $x$ will not be eliminated from $U$ since $x$ will not lose to any other item in $S_t$ during calls of SC.

We show that any other item in $U$ will be eliminated from $U$ after a finite number of iterations of the while loop in Probe-Max. Let $y \neq x$ be an item in $U$. Since $x$ is the maximum, $y \prec x$ in the true ranking. Whenever $\epsilon_{\tau_{y,x}} \leq \Delta_{x,y}$, a successful call of SC $\left(x, y, 2\delta/n^2, \tau_{x,y}\right)$ will return the result $x \succ y$ and remove $y$ from $U$ if $\mathcal{E}^{bad}$ does not happen. Since $\epsilon_{\tau_{y,x}}$ converges to 0, there must exist $\tau^*_{x,y}$ such that $\epsilon_{\tau^*_{x,y}} \leq \Delta_{x,y}$. After each execution of SC, the corresponding $\tau$ value increases by one, therefore after at most $\binom{n}{2}\tau^*_{x,y}$ iterations of the while loop, SC $\left(x, y, 2\delta/n^2, \tau^*_{x,y}\right)$ must have been called. The same argument holds for any $y \in U, y \neq x$.

**Sample complexity.** We first note the asymptotic behavior that for any $N > 0$,

$$\sum_{\tau=1}^{N} b_\tau \leq \sum_{\tau=1}^{N} \frac{2}{4^{-\tau}} \log \frac{c\tau^2 n^2}{\delta} \leq \sum_{\tau=1}^{N} \frac{2}{4^{-\tau}} \log \frac{cN^2 n^2}{\delta} = O\left(4^N \log \frac{cN^2\delta^2}{\delta}\right) = O\left(b_N\right). \quad (19)$$

Without loss of generality, we assume the true ranking is $1 \succ 2 \succ \cdots \succ n$. When $\mathcal{E}^{bad}$ does not happen, all comparison results returned by SC coincide with the true ranking. Therefore, for every $i \in [n-1]$, item $i$ belongs to $S_1, S_2, \ldots, S_i$ and gets eliminated during the execution of Probe-Max$\left(S_i, 2\delta/n^2\right)$.

Recall that SC is only called over item pairs in which at least one of them is a maximal element. For every SC called on items $a, b$, if $a$ is maximal, we say item $a$ initializes the comparison and we charge the number of comparisons taken by SC to item $a$ (if both $a$ and $b$ are maximal, we charge the number of samples to both). Let $c(a)$ denote the number of comparisons charged to $a$. The total sample complexity of Probe-Rank is thus at most $\sum_{a \in [n]} c(a)$.

Fix $i \in [n]$. We use $\tau_i^\circ$ to denote the value of $\tau_{i,i-1}$ when the order between $i$ and $i-1$ is revealed. Define $\tau_1^\circ = 0$ for completeness. We note that the order between $i$ and $i-1$ can not be inferred from any other comparison results therefore can only be returned by SC $\left(i, i-1, 2\delta/n^2, \tau_i^\circ\right)$. When $\mathcal{E}^{bad}$ does not happen, $\tau_i^\circ \leq \left\lceil \log \frac{1}{\Delta_{i,i-1}} \right\rceil$ since a successful call of SC $\left(i, i-1, 2\delta/n^2, \left\lceil \log \frac{1}{\Delta_{i,i-1}} \right\rceil\right)$ will return the order.

For each $j \neq i$, we use $\tau^*_{i,j}$ to denote the value of $\tau_{i,j}$ when the last time SC is initialized by $i$ and called over $i, j$ before the beginning of Probe-Max$\left(S_i, 2\delta/n^2\right)$. In other words, for any $\tau > \tau^*_{i,j}$, if SC $\left(i, j, 2\delta/n^2, \tau\right)$ is called in Probe-Max$\left(S_t, 2\delta/n^2\right)$ for some $t < i$, then it must not be initialized by $i$. Moreover, we use $\tau^t_{i,j}$ to denote the value of $\tau_{i,j}$ right after Probe-Max$(S_t, 2\delta/n^2)$ terminates. Since $i$ is ranked and removed from $T$ after Probe-Max$(S_i, 2\delta/n^2)$ is called, $\tau^i_{i,j}$ is also the value of $\tau_{i,j}$ when Probe-Rank terminates. It is clear that

$$c(i) \leq \sum_{j \neq i} \sum_{\tau=1}^{\tau^*_{i,j}} b_\tau + \sum_{j \neq i} \sum_{\tau=\tau^{i-1}_{i,j}+1}^{\tau^i_{i,j}} b_\tau. \quad (20)$$

We consider the first term on the right-hand side of (20). Before Probe-Max$\left(S_{i-1}, 2\delta/n^2\right)$ terminates, item $i-1$ is in $T$. Therefore, whenever $i$ is a maximal element, the order between $i$ and $i-1$ must have not been revealed. So when $i$ initializes the comparison SC $\left(i, j, 2\delta/n^2, \tau^*_{i,j}\right)$, the item pair $(i, i-1)$ is also in the set of 'legitimate' pairs $P$. Therefore, $\tau^*_{i,j}$ is no larger than the value of $\tau_{i,i-1}$ at that point, and further no larger than $\tau_i^\circ$. The same argument holds for any $j$. It follows that

$$\sum_{j \neq i} \sum_{\tau=1}^{\tau^*_{i,j}} b_\tau \leq \sum_{j \neq i} \sum_{\tau=1}^{\tau^*_{i,j}} b_\tau \leq \sum_{\tau=1}^{\tau_i^\circ} nb_\tau. \quad (21)$$

Next, we bound the second term on the right-hand side of (20). Note that if there is no SC called during Probe-Max$(S_i, 2\delta/n^2)$, then $\sum_{j \neq i} \sum_{\tau=\tau_{i,j}^{i-1}+1}^{\tau_{i,j}^i} b_\tau = 0$. So it suffices to consider the case when at least one instance of SC is called during Probe-Max$(S_i, 2\delta/n^2)$. Consider the last group of SC called in Probe-Max$(S_i, 2\delta/n^2)$, here group means that there might be multiple item pairs whose $\tau$ values are the minimum in $P$. Denote their $\tau$ values by $\tau^i$. There must be some SC $\left(a_i, b_i, 2\delta/n^2, \tau^i\right)$ returning $b_i \succ a_i$ such that $a_i$ is a maximal item, otherwise no maximal item is removed from $U$ and Probe-Max will not terminate. When $\mathcal{E}^{bad}$ does not happen, $a_i$ is not the maximum in $S_i$ so $a_i > i$. Thus, item $a_i - 1$ is also in $S_i$ and before the call of SC $\left(a_i, b_i, 2\delta/n^2, \tau^i\right)$, the ordering between $a_i - 1$ and $a_i$ is not revealed, i.e., $\tau^i \leq \tau_{a_i}^\circ$. Moreover, $\tau_{i,j}^i \leq \tau^i$ by the fact that we always compare item pairs with the smallest $\tau$ values. It follows that

$$\sum_{j \neq i} \sum_{\tau=\tau_{i,j}^{i-1}+1}^{\tau_{i,j}^i} b_\tau \leq n \sum_{\tau=1}^{\tau^i} b_\tau = O\left(nb_{\tau^i}\right). \tag{22}$$

The same argument holds for all $i \in [n-1]$.

Consider the sets

$$\mathcal{D}_1 = \{b_{\tau^i} : i = 1, 2, \ldots, n-1\}, \quad \mathcal{D}_2 = \cup_{i=2}^n \mathcal{D}_2^i = \cup_{i=2}^n \{b_\tau : \tau = 1, 2, \ldots, \tau_i^\circ\}. \tag{23}$$

We claim that if $i_1 \neq i_2$, then the pairs $(a_{i_1}, \tau^{i_1})$ and $(a_{i_2}, \tau^{i_2})$ do not equal. With the facts that $a_i > i$ and $\tau^i \leq \tau_{a_i}^\circ$, there is an injective mapping from $\mathcal{D}_1$ to $\mathcal{D}_2$ given by $b_{\tau^i}$ is mapped to the element $b_{\tau^i}$ in $\mathcal{D}_2^{a_i}$. It follows that

$$\sum_{i=1}^{n-1} O\left(nb_{\tau^i}\right) = O\left(\sum_{x \in \mathcal{D}_1} nx\right) \leq O\left(\sum_{x \in \mathcal{D}_2} nx\right) = O\left(\sum_{i=2}^n \sum_{\tau=1}^{\tau_i^\circ} nb_\tau\right). \tag{24}$$

The reason for pairs $(a_{i_1}, \tau^{i_1})$ and $(a_{i_2}, \tau^{i_2})$ equal if and only if $i_1 = i_2$ is as follows. Let $i_2 > i_1$ and suppose $a_{i_1} = a_{i_2} = a$. When SC $\left(a, b_{i_1}, 2\delta/n^2, \tau^{i_1}\right)$ is called, SC $\left(a, b, 2\delta/n^2, \tau^{i_1}\right)$ for all $b$ such that $(a, b) \notin T$ and $\tau_{a,b} = \tau^{i_1}$ are also called. It follows that $\tau_{a,b} > \tau^i$ for all such $b$ after this point. When SC $\left(a, b_{i_2}, 2\delta/n^2, \tau^{i_2}\right)$ is called, the order between $a$ and $b_{i_2}$ is not know and thus also not known when SC $\left(a, b_{i_1}, 2\delta/n^2, \tau^{i_1}\right)$ was called. So $\tau^{i_2}$ must be larger than $\tau^{i_1}$.

Combining (20), (21) and (24) gives,

$$\sum_{i=1}^n c(i) \leq \sum_{i=2}^n \sum_{j \neq i} \sum_{\tau=1}^{\tau_{i,j}^*} b_\tau + \sum_{i=1}^{n-1} \sum_{j \neq i} \sum_{\tau=\tau_{i,j}^{i-1}+1}^{\tau_{i,j}^i} b_\tau \tag{25}$$

$$\leq O\left(\sum_{i=2}^n \sum_\tau^{\tau_i^\circ} nb_\tau\right) = O\left(n \sum_{i=2}^n b_{\tau_i^\circ}\right). \tag{26}$$

The desired sample complexity follows from $\tau_i^\circ \leq \left\lceil \log \frac{1}{\Delta_{i,i-1}} \right\rceil$ and

$$b_{\lceil \log \frac{1}{\Delta} \rceil} = O\left(\frac{1}{\Delta^2}\left(\log \log \frac{1}{\Delta} + \log \frac{n}{\delta}\right)\right), \tag{27}$$

which completes the proof. $\qquad \square$

## B   A sample-efficient variant of Probe-Rank

In this section, we present a variant of Probe-Rank, named *Probe-Rank-SE*. When demonstrating more detailed experiments in Appendix C, Probe-Rank-SE is also included and is shown to have better practical performance. However, we will not prove its correctness due to the high similarity it shares with Probe-Rank.

Compared with Probe-Rank, the variant Probe-Rank-SE finds the ranking also by performing $n - 1$ steps of maxing and differs only in the subroutine for collecting comparison samples. Specifically,

Probe-Rank-SE takes queries from all unknown item pairs simultaneously. Comparison results for pairs that terminate earlier are still collected and stored in the graph $T$, which represents our current knowledge about the ranking. We use $T$ to decide whether to pause, drop or resume comparisons of remaining item pairs.

We adopt the Successive Elimination (SE) algorithm from [9], shown in Algorithm 4, as a procedure to perform comparisons.

---

**Subroutine 4** Successive Elimination (modified for comparing two items)

---

1: **Input:** items $i, j$, confidence level $\delta$
2: $t = 1$;
3: **while** true **do**
4:     Compare $i$ and $j$ for $2^t$ times; Let $\widehat{p}_i^t$ be the winning rate of $i$;
5:     Let $\alpha_t = \sqrt{\frac{\log(ct^2/\delta)}{2^t}}$, $c = \frac{\pi^2}{3}$;
6:     **return** $i \succ j$ **if** $\widehat{p}_i^t - \frac{1}{2} > \alpha_t$; **return** $j \succ i$ **if** $\widehat{p}_i^t - \frac{1}{2} < -\alpha_t$; $t = t + 1$ **else**;
7: **end while**

---

It was shown that with probability at least $1 - \delta$, Subroutine 4 correctly returns the more preferred item between $i$ and $j$ using at most $O\left(\frac{1}{\Delta_{i,j}^2}\left(\log\frac{1}{\delta} + \log\log\frac{1}{\Delta_{i,j}}\right)\right)$ comparisons [9, Remark 1].

In Probe-Rank-SE, we do not call SE directly, rather, SE is used as a black-boxed unit that repeatedly collects query samples from the input pair $i, j$. Moreover, after every sample, it generates feedback which is either Null, $i \succ j$ or $j \succ i$, where Null corresponds to that the number of samples has not accumulated to $2^t$ or $\left|\widehat{p}_i^t - \frac{1}{2}\right| < \alpha_t$; feedback $i \succ j$ and $j \succ i$ correspond to that inside the black box, SE actually terminates and returns the order between $i$ and $j$. Note that the SE procedure can be replaced by any algorithm that can rank two items, including all best-arm-identification algorithms.

Denote the instance of Successive Elimination that runs over items $i, j$ with confidence level $\delta$ as $\text{SE}_{i,j}(\delta)$. When the value of $\delta$ is given without ambiguity, we will drop the dependence and write $\text{SE}_{i,j}$ as a shorthand. We define two operations on $\text{SE}_{i,j}$, named advance and feed. The advance operation returns one of the three possible internal outcomes, Null, $i \succ j$ or $j \succ i$. The feed operation is used for simulating the sampling process. We write $\text{feed}(\text{SE}_{i,j}, Y_{i,j})$ to represent that $\text{SE}_{i,j}$ is fed with a comparison sample $Y_{i,j}$. As a black-boxed unit, before advance returns one of $i \succ j$ and $j \succ i$, advance and feed operations are invoked in an alternating fashion. The idea of viewing a sampling subroutine as a black-box controlled by artificial operations was also used in [1], but for a different problem setting.

Probe-Rank-SE is presented in Algorithm 5. We initialize $\binom{n}{2}$ independent instances of $\text{SE}_{i,j}\left(2\delta/n^2\right)$, each for obtaining the order between an item pair $(i, j), 1 \leq i < j \leq n$. The probability of being unable to recover the true ranking is thus upper bounded by probability that at least one of the SE instances fails, which is at most $\delta$. Same as Probe-Rank, we use $T$ to denote the transitive closure composed of results returned by the SE instances.

---

**Algorithm 5** Probe-Rank-SE

---

1: **Input:** items $[n]$, confidence level $\delta$
2: $S_1 = [n]$, $Ans = [0]^n$; initialize $T$;
3: initialize $\text{SE}_{i,j}\left(2\delta/n^2\right)$ for all $1 \leq i < j \leq n$;
4: **for** t from 1 to $n - 1$ **do**
5:     $i_{max} = $ Probe-Max-SE$(S_t)$;
6:     remove $i_{max}$ from $T$; $Ans[t - 1] = i_{max}$; $S_{t+1} = S_t \setminus \{i_{max}\}$;
7: **end for**
8: $Ans[n - 1] = S_n[0]$; **return** $Ans$;

---

The procedure Probe-Max-SE serves as a switch for the SE instances. Let $S_t^2$ denote the set of unordered item pairs $\{(i, j) : i, j \in S_t, i \neq j\}$. In each round $t$, all SE instances for 'legitimate' pairs in $S_t^2$ are turned on and take queries in a round-robin fashion. 'Legitimate' pairs are similarly

defined as in Probe-Rank. A pair $(i, j)$ is 'legitimate' if the order between $i, j$ is unknown, i.e., not in $T$, and at least one of $i$ and $j$ is a maximal element in $S_t$.

---

**Algorithm 6** Probe-Max-SE($S_t$)

---

 1: Let $U$ be sets of maximal elements according to $T$
 2: **while** $|U| \geq 1$ **do**
 3:     $C = [\,]$
 4:     **for** $(i, j)$ in $S_t^2$ **do**
 5:         **if** $(i \in U$ or $j \in U)$ and $(i, j) \notin T$ **then**
 6:             compare $i$ with $j$ once and get result $Y_{i,j}$; feed $(\text{SE}_{i,j}(\delta/n^2), Y_{i,j})$
 7:             **if** advance $(\text{SE}_{i,j}(2\delta/n^2)) == i \succ j$ **then** $C$.append($[i, j]$);
 8:             **else if** advance $(\text{SE}_{i,j}(2\delta/n^2)) == j \succ i$ **then** $C$.append($[j, i]$);
 9:             **end if**
10:         **end if**
11:     **end for**
12:     **for** $w, l$ in $C$ **do**
13:         **if** $(w, l) \notin T$ **then**
14:             $T = T \cup (w \succ l)$;
15:             **if** $|U| > 1$ and $l \in U$ **then** $U = U \setminus \{l\}$;
16:             **end if**
17:         **end if**
18:     **end for**
19: **end while**
20: **return** $U[0]$;

---

# C   Additional experiments

In this section, we present more detailed numerical experiments comparing the sample complexities of Probe-Rank, Probe-Rank-SE and the state-of-the-art algorithm IIR by Ren et al. [25]. In particular, we focus on the `WST`, `SST`, `NON-SST` and `ADJ-ASYM` settings and perform these three algorithms with various parameters. Same as the results presented in Section 6, all experiments are averaged over 100 independent trials. For each trial, the ground truth ranking $\sigma$ is generated uniformly at random and the comparison probabilities are assigned according to the chosen setting. The confidence level $\delta$ is fixed to be $0.1$. Throughout the experiment, every trial for every algorithm successfully recovered the correct ranking. Moreover, for IIR, if the rank has not been recovered after the sample complexity reaches $10^9$, we manually stop the ranking process and record the sample complexity as $10^9$ to avoid extremely large running times. Note that the extreme cases happen in Figures 8(a), 8(b) 8(c) and 12(d).

Figures 5, 6, 7 and 8 compare the three algorithms under different settings where the difficulty parameter $\Delta_d$ is fixed and the number of items $n$ varies from 10 to 100. Figures 9, 10, 11 and 12 compare the three algorithms under different settings where the number of items $n$ is fixed and the difficulty parameter $\Delta_d$ varies from 0.1 to 0.4. It can be seen that Probe-Rank and its variant always consume less samples than IIR to recover the true ranking. Note that in the WST setting, comparison probabilities are all identically distributed and thus on average, adjacent items are as hard as nonadjacent items to compare. When $\Delta_d$ is fixed, as $n$ gets larger and larger, IIR will eventually outperform Probe-Rank. This is consistent with our theoretical results presented in Section 5. Moreover, as indicated by the experimental results, Probe-Rank-SE can further reduce the sample complexity compared with Probe-Rank.

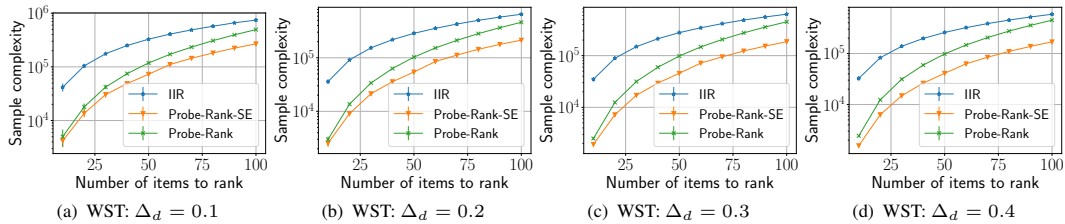

Figure 5: Comparison of Probe-Rank, Probe-Rank-SE and IIR under the WST setting. In each subfigure, $\Delta_d$ is fixed while the number of items varies.

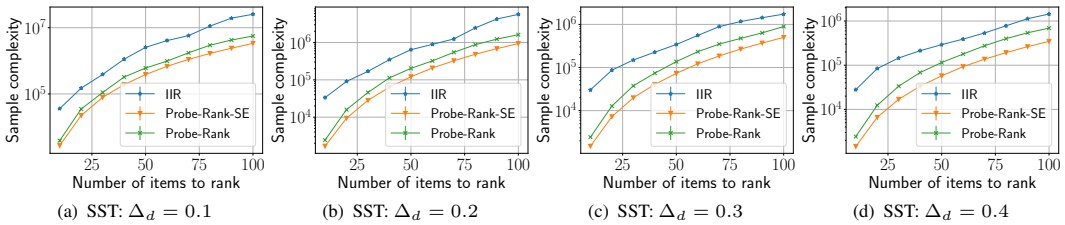

Figure 6: Comparison of Probe-Rank, Probe-Rank-SE and IIR under the SST setting. In each subfigure, $\Delta_d$ is fixed while the number of items varies.

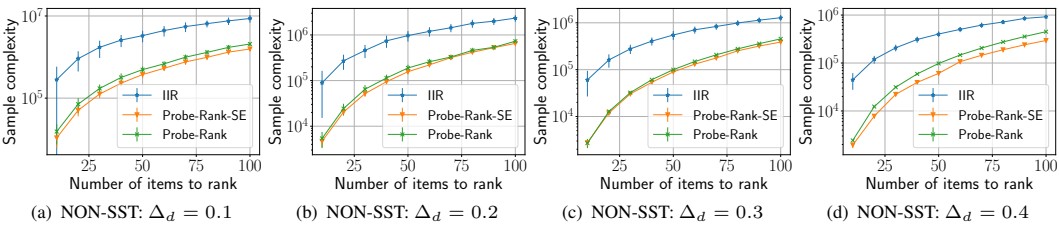

Figure 7: Comparison of Probe-Rank, Probe-Rank-SE and IIR under the NON-SST setting. In each subfigure, $\Delta_d$ is fixed while the number of items varies.

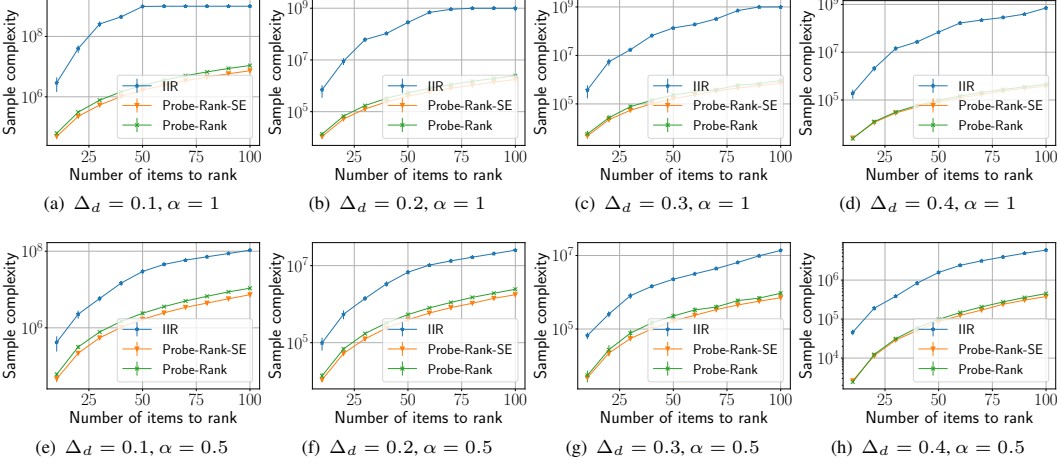

Figure 8: Comparison of Probe-Rank, Probe-Rank-SE and IIR under the ADJ-ASYM setting. In each subfigure, $\Delta_d$ and $\alpha$ are fixed while the number of items varies.

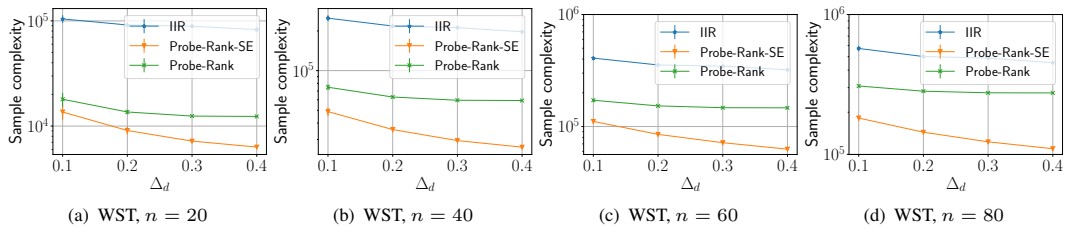

Figure 9: Comparison of Probe-Rank, Probe-Rank-SE and IIR under the WST setting. In each subfigure, $n$ is fixed while $\Delta_d$ varies.

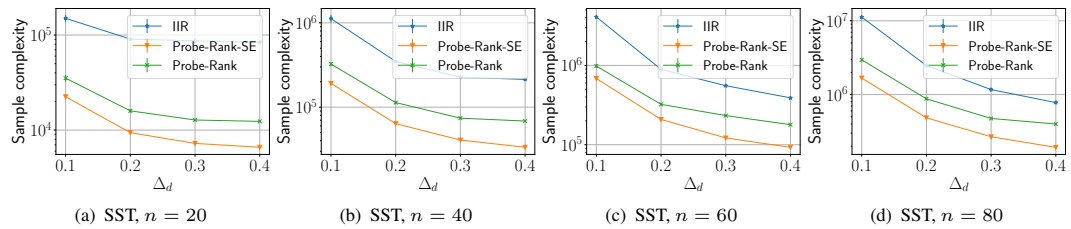

Figure 10: Comparison of Probe-Rank, Probe-Rank-SE and IIR under the SST setting. In each subfigure, $n$ is fixed while $\Delta_d$ varies.

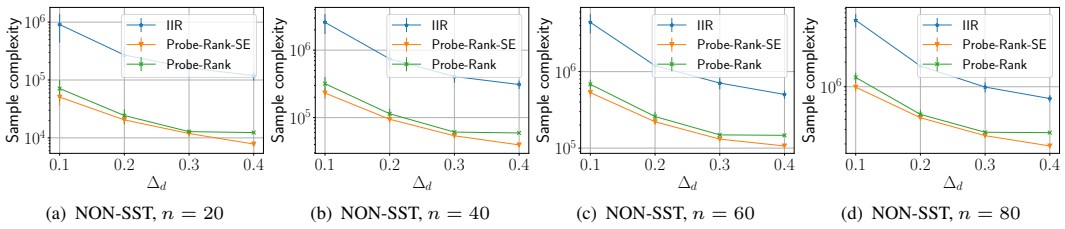

Figure 11: Comparison of Probe-Rank, Probe-Rank-SE and IIR under the NON-SST setting. In each subfigure, $n$ is fixed while $\Delta_d$ varies.

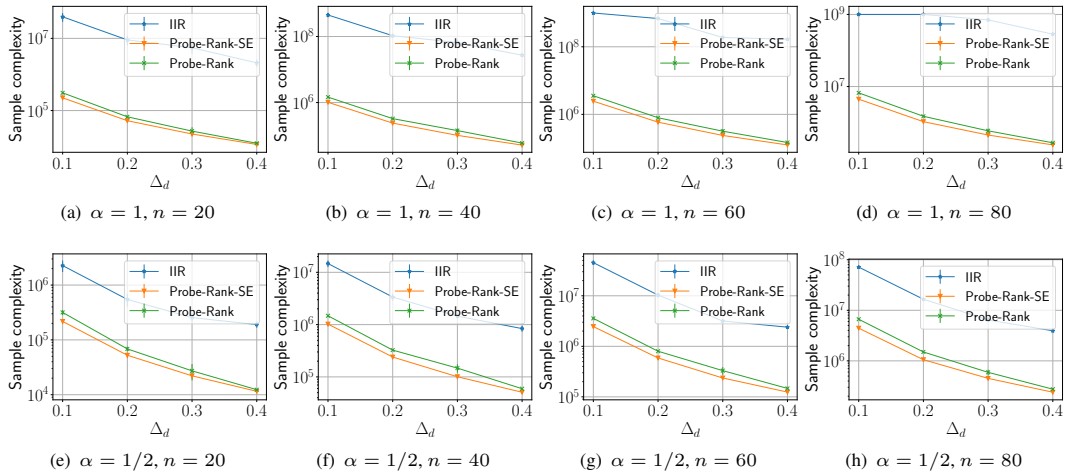

Figure 12: Comparison of Probe-Rank, Probe-Rank-SE and IIR under the ADJ-ASYM setting. In each subfigure, $n$ and $\alpha$ are fixed while $\Delta_d$ varies.

# D  Lower bound analysis

In this section, we present the reduction from the maxing problem for $\mathcal{I}_{SNG}$ (Problem 2) to the maxing problem for $\mathcal{I}_{WST}$ (Problem 1). The two problems are restated as follows.

We first consider another problem instance $\mathcal{I}_{SYM}$ and show that the maxing problem for $\mathcal{I}_{SYM}$ can be reduced to the maxing problem for $\mathcal{I}_{WST}$. The problem instance $\mathcal{I}_{SYM}$ is modified from $\mathcal{I}_{WST}$ by setting the values of $p_{i,j}$ to be exactly $\frac{1}{2}$.

**Problem 3** ($\mathcal{I}_{SYM}$). Consider $n$ items with an underlying ordering '$\succ$'. The comparison probabilities are defined as:

$$p_{i,j} := \begin{cases} \frac{1}{2} + \Delta, & \text{if } i \succ j \text{ and } i, j \text{ are adjacent,} \\ \frac{1}{2} - \Delta, & \text{if } i \prec j \text{ and } i, j \text{ are adjacent,} \\ \frac{1}{2}, & \text{otherwise.} \end{cases}$$

Note that $\mathcal{I}_{SYM}$ does not satisfy the WST condition as the comparison probabilities can be $\frac{1}{2}$. However, any $\delta$-correct algorithm that finds the maximum item for $\mathcal{I}_{WST}$ efficiently can also find the maximum item for $\mathcal{I}_{SYM}$ efficiently, shown as follows.

**Reduction from $\mathcal{I}_{SYM}$ to $\mathcal{I}_{WST}$**    Let $\mathcal{A}$ be any $\delta$-algorithm that finds the maximum item for any instance that satisfies the WST condition. Algorithm $\mathcal{A}$ is also able to find the maximum item for $\mathcal{I}_{WST}$ with any $c > 0$. Consider any interaction trajectory $\mathcal{T}$ defined by the sequence of comparisons (including the choices for item pairs and their outcomes) with length smaller than $Cn^2 \log(1/\delta)/\Delta^2$ for some constant $C$. Under the two instances $\mathcal{I}_{SYM}$ and $\mathcal{I}_{WST}$, the probabilities of occurrences of $\mathcal{T}$, denoted $\mathbb{P}_{SYM}$ and $\mathbb{P}_{WST}$, satisfy

$$\begin{aligned}
\frac{\mathbb{P}_{SYM}(\mathcal{T})}{\mathbb{P}_{WST}(\mathcal{T})} &\geq \left( \frac{1/2 - cn^{-10}\Delta^2/\log(1/\delta)}{1/2} \right)^{Cn^2 \log(1/\delta)/\Delta^2} \\
&> \left( \frac{1}{1 + 4cn^{-10}\Delta^2/\log(1/\delta)} \right)^{Cn^2 \log(1/\delta)/\Delta^2} \\
&\geq e^{-4cn^{-10}\Delta^2/\log(1/\delta) \cdot Cn^2 \log(1/\delta)/\Delta^2} \\
&= e^{-4cCn^{-8}} \\
&\geq 1 - \delta,
\end{aligned} \tag{28}$$

where the first inequality holds because the likelihood ratio for one query is upper bounded by the base and the number of queries on nonadjacent pairs is bounded by the exponent; the second inequality assumes $cn^{-10}\Delta^2/\log(1/\delta) < 1/4$, which holds for $n$ sufficiently large; the third inequality is due to $(1 + x) \leq e^x$. The last inequality can hold by choosing a small enough $c$.

If $\mathcal{A}$ solves the maxing problem for $\mathcal{I}_{WST}$ with probability at least $1 - \delta$ and conducts at most $Cn^2 \log(1/\delta)/\Delta^2$ comparisons, then by inequality (28),

$$\begin{aligned}
\mathbb{P}_{SYM}(\mathcal{A} \text{ finds the correct maximum}) = \sum_{\mathcal{T} \in \mathcal{E}_f} \mathbb{P}_{SYM}(\mathcal{T}) &\geq (1 - \delta) \sum_{\mathcal{T} \in \mathcal{E}_f} \mathbb{P}_{WST}(\mathcal{T}) \\
&= (1 - \delta)\, \mathbb{P}_{WST}(\mathcal{A} \text{ finds the correct maximum}) \\
&\geq (1 - \delta)^2 \\
&> 1 - 2\delta,
\end{aligned} \tag{29}$$

where $\mathcal{E}_f$ denotes the collection of trajectories where $\mathcal{A}$ returns the correct maximum. In other words, $\mathcal{A}$ is also a $2\delta$-correct algorithm that solves the maxing problem for $\mathcal{I}_{SYM}$ and conducts at most $Cn^2 \log(1/\delta)/\Delta^2$ comparisons. Further, if we force $\mathcal{A}$ to terminate after $Cn^2 \log(1/\delta)/\Delta^2$ comparisons have been made, then $\mathcal{A}$ is still correct with probability at least $1 - \delta$ and with expected number of samples upper bounded by $Cn^2 \log(1/\delta)/\Delta^2$.

The second step is to reduce the maxing problem for $\mathcal{I}_{SNG}$ to the maxing problem for $\mathcal{I}_{SYM}$.

**Reduction from $\mathcal{I}_{SNG}$ to $\mathcal{I}_{SYM}$**    Let $\mathcal{A}$ be any $\delta$-correct algorithm that can find the maximum item for $\mathcal{I}_{SYM}$. Without loss of generality, we can assume that when comparing an item pair $i, j$, $i < j$,

$\mathcal{A}$ takes in answer 1 representing $i$ is more preferred and answer 0 representing $j$ is more preferred. We construct a $\delta$-correct algorithm $\mathcal{A}'$ that can find the maximum item for $\mathcal{I}_{SNG}$ from $\mathcal{A}$:

> Given $\mathcal{A}$, whenever $\mathcal{A}$ compares item pair $(i, j)$,
> with probability $1/2$, $\mathcal{A}'$ queries 'if $i \succ j$', gets the sample $Y$ and feeds $Y$ to $\mathcal{A}$;
> with probability $1/2$, $\mathcal{A}'$ queries 'if $j \succ i$', gets the sample $Y$ and feeds $1 - Y$ to $\mathcal{A}$. Whenever $\mathcal{A}$ terminates and return an item, $\mathcal{A}'$ also terminates and return the same item.

It is clear that, if $\mathcal{A}$ queries an adjacent pair $i, j$ with $i \succ j$, the feedback $Y$ is an average over $Y_{i,j}$ (Ber($\frac{1}{2}$)) and $1 - Y_{j,i}$ (1-Ber($\frac{1}{2} - 2\Delta$)), which is Ber($\frac{1}{2} + \Delta$); if $i, j$ are adjacent and $i \prec j$, the feedback $Y$ is an average of Ber($\frac{1}{2} - 2\Delta$) and $1 -$Ber($\frac{1}{2}$), which is Ber($\frac{1}{2} - \Delta$); if $i, j$ are nonadjacent, the feedback $Y$ is an average of two Ber($\frac{1}{2}$) random variables, which is still Ber($\frac{1}{2}$). Therefore, $\mathcal{A}$ gets the same feedback when it is performed over $\mathcal{I}_{SYM}$. If $\mathcal{A}$ is a $\delta$-correct maxing algorithm for $\mathcal{I}_{SYM}$ and conducts at most $Cn^2 \log(1/\delta)/\Delta^2$ comparisons on average, then $\mathcal{A}'$ is a $\delta$-correct maxing algorithm for $\mathcal{I}_{SNG}$ with the same sample complexity.

To summarize, if there exists a $\delta$-correct algorithm $\mathcal{A}$ that solves the maxing problem for $\mathcal{I}_{WST}$ and conducts at most $Cn^2 \log(1/\delta)/\Delta^2$ on average, then with the reduction, we can conclude there exists a $2\delta$-correct algorithm $\mathcal{A}'$ that solves Problem 2 with the same sample complexity. Since the above argument holds for any $C > 0$ and in Section 7, we argued that Example 2 requires $\Omega(n^2 \log(1/\delta)/\Delta^2)$ queries, we thus conjecture that any $\delta$-correct algorithm $\mathcal{A}$ that solves the maxing (and thus ranking) problem for $\mathcal{I}_{WST}$ conducts $\Omega(n^2 \log(1/\delta)/\Delta^2)$ comparisons.