# OpenReview forum: "Active Ranking without Strong Stochastic Transitivity"
_NeurIPS.cc/2022/Conference — NeurIPS 2022 Accept_

### Official Review · Reviewer_XjGa · 2022-07-08

**Rating:** 6
**Confidence:** 3
**Soundness:** 4 excellent
**Presentation:** 3 good
**Contribution:** 3 good

**Summary:**

The paper analyzes the problem to identify the underlying ranking in dueling bandits under the assumption that the underlying preference probabilities fulfill weak stochastic transitivity.
The authors introduce a novel algorithm that is able to solve the problem in a $\delta$-PAC manner and comes with an instance-wise sample complexity guarantee that depends only on the gaps of those arms, which are adjacent according to the underlying ranking, and not on the other gaps.

**Questions:**

- Thm. 1 in [3] shows that maxing (and thus in particular ranking) in an $(\epsilon,\delta)$-PAC manner under WST requires in the worst-case $\Omega(n^{2})$ samples.  As their worst-case instance is similar to yours, you could potentially adapt their proof for proofing your conjecture. Did you try this?
- Thm 2: From looking at its proof, it appears that you could slightly strengthen Thm.2 and provide a more detailed bound without O-notation, i.e., you could explicitly provide the constant that is hidden in the O-term. Maybe you could state this constant for the sake of completeness in the appendix?
- Under SST $\Delta_i = \widetilde{\Delta}_i$  holds. You also mentioned RST in Sec. 3, and [3] e.g. discuss moderate stochastic transitivity. Could you say how $\Delta_i$ and $\tilde{\Delta}_i$ can differ under such transitivity assumptions? I.e., would Probe-Rank still be better than IIR in special cases under such assumptions?

[3] M. Falahatgar et al., The Limits of Maxing, Ranking, and Preference Learning, ICML 2018.

**Limitations:**

There do not seem to be any potential negative societal impacts of this work. The limitations of their theoretical results have adequately been addressed by the authors.

**Strengths And Weaknesses:**

Strengths:
1. The problem of ranking in dueling bandits is relevant for practical purposes and has already been studied in the literature. The algorithmic solution of the authors appears novel and apparently outperforms the current state-of-the-art methods in special cases, both empirically and in theory.
2. The authors prove an interesting upper bound on the sample complexity for Ranking under WST: In contrast to existing ones, theirs depends only on the gaps between adjacent arms (according to the ranking) and not on all gaps.
3. The paper is well-written, the presentation of the algorithm is easily understandable and well illustrated with an example. The theoretical results appear reasonable. I did not check all the proofs in full detail, but at first sight they seem to be correct.

Weaknesses:
1. In the appendix, the authors present with Probe-Rank-SE an improved variant of their algorithm, which they do not mention in the main paper. In my opinion, the authors should either focus on this variant or at least mention it in the main part.
2. The authors only conjecture (and not prove) a lower bound that would show (log-)optimality of their result. Even though the authors discuss where this conjecture comes from, the final step still needs to be made. And I have my doubts that this final step is possible, as one would have to add up expected sample complexity lower bounds that are valid w.r.t. different instances. However, this might suffice as motivation for the conjecture.

Minor Weaknesses and Typos:
- 29: It should be ''$p_{i,k} \geq \max (p_{i,j},p_{j,k}) > 1/2$''.
- 42: You say you focus on pairwise comparisons, and before you have already said that you focus on WST. As far as I know, WST is only defined for pairwise comparisons. Maybe you could reformulate this.
- 115: and [] usually
- 118: In [the] PAC settings
- 140: the comparison should last
- 2 in Alg.1: $b_{\tau} = \lceil ... \rceil$? Then, adapt (13) apparantly.
- 196: The lower bound of order $\tilde{\Delta}^{-2} \log(n/\delta)$ is not stated in Sec. 2, but only in footnote 1. Maybe, you could also mention it next to (2).
- 210: Theorem 2 does not show tightness of its bound, it does not contain a lower bound result.
- 217: addition[al] factor
- 225: Why $\Delta_{d}$, what does $d$ stand for?
- 318: under[r]
- 466: $\Delta_{i,j}>\epsilon_{\tau}$
- 579: that [] at least
- 3 in Alg.5: SE italic
- 612: (Problem 3 in the main paper) = (Probl. 1 in the appendix)
- 617: Sometimes in this section, WST etc. in the index is italic, sometimes not.

Additional Remark:

In Sec.1 you mention that SST may be too strong in some scenarios and motivate this way the usage of WST. This naturally raises the question how strong the WST assumption itself is.
Apart from [1], which detected violations of the Condorcet winner assumption (i.e., in particular of the WST assumption) in real-world examples and thus focused on identification of Copeland winners, [2] recently discussed testing whether the WST assumption is fulfilled or not. The authors showed that WST testing requires $\Omega(n^2 \Delta^{-2} \log (1/\delta))$ samples and, interestingly, they proved an instance-wise lower bound for corresponding solutions that depends on $\Omega(n^2)$ of the gaps and in particular not only on adjacent ones as your upper bound.

[1] $:=$ [30] in your main paper

[2] B. Haddenhorst et al., On testing transitivity in online preference learning. Machine Learning, 2021

---

> ### Author Response · Authors · 2022-08-02
> **Response to Reviewer XjGa**
>
> ## Reviewer XjGa
>
> We thank the reviewer for the positive feedback and for pointing out typos and other issues. We have revised our paper accordingly. The changes are marked blue in the revised manuscript. In the following, we address all comments in detail.
>
> ---
> **Q1:** "In my opinion, the authors should either focus on this variant or at least mention it in the main part."
>
> **A1:** We thank the reviewer for raising this point. We did not include the  Probe-Rank-SE variant in the main paper mainly due to the fact that it does not have a theoretical guarantee even though it empirically performs better in many cases. We have commented on the variant in the introduction section as suggested and will also be working on deriving a theoretical bound for Probe-Rank-SE in the future work.
>
> ---
> **Q2:** "The authors only conjecture (and not prove) a lower bound that would show (log-)optimality of their result. Even though the authors discuss where this conjecture comes from, the final step still needs to be made. And I have my doubts that this final step is possible, as one would have to add up expected sample complexity lower bounds that are valid w.r.t. different instances."
>
> **A2:** We thank the reviewer for the careful examination of our conjecture. We agree with the reviewer that the final step of adding up all instance lower bounds cannot lead to a rigorous proof. We intended the argument to serve as strong evidence for the conjecture. We are planning to establish a rigorous proof via information-theoretical techniques.
>
> ---
> **Q3:** "line 225: Why $\Delta_d$? what does $d$ stand for?"
>
> **A3:** $\Delta_d$ is an ad hoc notation that we introduce merely for our experiment. By choosing different values of $\Delta_d$, we get different problem instances for our algorithm to perform on. The letter $d$ is rather an arbitrary choice. It can indeed cause confusion for readers. However, as $\Delta_d$ appears frequently in our plots, we choose not to make changes at this point. We will replace it by perhaps $\Delta'$ in the final version. We thank the reviewer for pointing it out.
>
> ---
> **Q4:** "...This naturally raises the question how strong the WST assumption itself is. ...\[2\] recently discussed testing whether the WST assumption is fulfilled or not. The authors showed that WST testing requires $\Omega\left(n^2 \Delta^{-2}\log\left(1/\delta\right)\right)$ samples and, interestingly, they proved an instance-wise lower bound for corresponding solutions that depends on $\Omega\left(n^2\right)$ of the gaps and in particular not only on adjacent ones as your upper bound."
>
> **A4:** The validity of WST is indeed an interesting question. If we assume there exists a ranking over the set of items and that (noisy) comparisons are consistent with this ranking, then WST is satisfied. We believe WST can be considered a natural and reasonably weak assumption. However, we agree that there are situations that WST does not hold as a ranking over items may not exist or, if it does, all comparison probabilities are not necessarily consistent with that ranking. We thank the reviewer for pointing out relevant references to testing WST and ranking when it does not hold. We will add a discussion of WST to the paper and will discuss the related work. The problem of testing for WST seems to bear a similarity to the ranking problem, although it is not clear if results from one can be transferred to the other problem, a relationship that we plan to study in the future.
>
> ---
> **Q5:** "Did you try adapting the proof of Thm. 1 in \[Falahatgar2018\]?"
>
> **A5:** We thank the reviewer for the suggestion. Indeed, we examined the proof in \[Falahatgar2018\] in hope of adapting their technique to prove our instance-dependent lower bound. \[Falahatgar2018\] assumes deterministic outcomes from comparison, and thus each pair has to be queried at most only once. Then essentially the pigeon-hole theorem guarantees each item must be queried $\Omega(n)$ times to fully identify the exact ranking. To derive gap-dependent lower bound, now each pair has to be queried for an indefinite number of times. That is why their combinatorial argument cannot be immediately adapted to our lower bound.
>
> Attempts were made to incorporate their proof with the generalized sequential probability ratio test. We tried to maintain the probability ratio test for each pair, and conclude on the pair that triggers the test. In this way, it is possible to analyze as if the pairs have deterministic comparisons. Unfortunately, incorporating the ratio test drastically complicate the proof and we were not able to work it out within a short time. It is still unclear whether the information-theoretical argument can be combined with their proof idea.

---

> > ### Author Response · Authors · 2022-08-02
> > **Response to Reviewer XjGa (continued)**
> >
> > ---
> > **Q6:** "From looking at its proof, it appears that you could slightly strengthen Thm.2 and provide a more detailed bound without O-notation, i.e., you could explicitly provide the constant that is hidden in the O-term. Maybe you could state this constant for the sake of completeness in the appendix?"
> >
> > **A6:** We thank the reviewer for this extremely helpful suggestion. We agree that by computing explicit coefficients in each step of the proof of Thm. 2, we could get an exact upper bound and make the theorem stronger. We will be working on it and hope to get the result as soon as possible.
> >
> > ---
> > **Q7:** "Could you say how $\Delta_i$ and $\tilde\Delta_i$ can differ under such transitivity assumptions? I.e., would Probe-Rank still be better than IIR in special cases under such assumptions?"
> >
> > **A7:** RST assumes that for all $i\succ j\succ k, \Delta_{i,k} \ge \gamma\max\left( \Delta_{i,j},\Delta_{j,k} \right)$ for some $0<\gamma<1$. Fix $i$, for any $j$, $\Delta_{i,j} \ge \gamma\min\left( \Delta_{i,i-1},\Delta_{i,i+1} \right) = \gamma\tilde\Delta_i$. It follows that $\gamma\tilde\Delta_i\le \Delta_i\le\tilde\Delta_i$, which means that $\Delta_i$ is still not larger than $\tilde\Delta_i$, but instead of having a lower bound 0 (as in WST), it has a larger lower bound $\gamma\tilde\Delta_i$.
> >
> > Comparing Probe-Rank to IIR, since $\Delta_i$ can still be strictly less than $\tilde\Delta_i$, Probe-Rank is possible to outperform IIR. The smaller $\gamma$ is, the more advantage Probe-Rank has. Specifically, the sample complexity upper bound for IIR is $\tilde O\big( \sum_{i}\frac{1}{\Delta_i^2} \big)$ and the sample complexity upper bound for Probe-Rank is $\tilde O\big( \sum_{i}\frac{n}{\tilde\Delta_i^2} \big)$. By replacing $\Delta_i$ with $\gamma\tilde\Delta_i$, the two sample complexities thus differ by a factor of $\frac{n}{\gamma^2}$. So if $\gamma<\frac{1}{\sqrt{n}}$, then Probe-Rank has a better theoretical guarantee. However, if $\gamma$ remains a constant, then as $n$ increases, IIR will eventually outperform Probe-Rank.

---

### Official Review · Reviewer_xP6H · 2022-07-10

**Rating:** 7
**Confidence:** 4
**Soundness:** 3 good
**Presentation:** 3 good
**Contribution:** 3 good

**Summary:**

This work designs an active ranking algorithm for WST. Authors also analyze the sample complexity and correctness of the algorithm. They propose conjecture on lower bound which if true implies the near optimality of their algorithm. They also provide numerical experiments contrasting their algorithm against SOTA algorithms under more restrictive settings and show pros and cons.

**Questions:**

I think it might be better to present some examples where this algorithm is necessary and epsilon-ranking isn't sufficient.

**Limitations:**

Authors have adequately addressed the limitations

**Strengths And Weaknesses:**

Strengths:
The sample complexity analysis is quite unique and clever. They divide the complexity analysis into two parts and bound each part using simple arguments. This makes following the proof quite easy.


Weakness:
Even though motivation for WST is justified well, the authors didn't mention why one needs an ordering stronger than epsilon-ranking.
Their algorithm has complexity of O(n^2) which is essentially same as that of the trivial epsilon-ranking. Having said that it is important to establish bounds for exact ranking and hence I lean towards accepting the paper.

---

> ### Author Response · Authors · 2022-08-02
> **Response to Reviewer xP6H**
>
> ## Reviewer xP6H
>
> We thank the reviewer for the positive feedback and for bringing up the point of exact ranking being not very well motivated.
>
> We would like to comment that an exact ranking is preferred over an epsilon-ranking in competitive applications like voting/sport games, where people are not satisfied with an approximate result. Furthermore, as suggested by \[Ren2019\], analyzing the exact ranking helps us to gain a better understanding about the instance-wise upper and lower bounds while epsilon-ranking only focuses on the worst-case scenario. So in applications where the actual probability gaps are large, exact algorithms might achieve lower sample complexities.
>
> We have revised our paper to provide motivation and an example as suggested (see Section 3). The changes are marked blue in the revised manuscript.
>
> ---
> **Q1:** "Their algorithm has complexity of $O(n^2)$ which is essentially same as that of the trivial epsilon-ranking."
>
> **A1:** While both our algorithm and the trivial epsilon-ranking algorithm have an $O\left(n^2 \right)$ factor in the sample complexity, our bound is instance-wise and depends on the actual probability gaps. In contrast, the bound for epsilon-ranking algorithm has an dependency on $\epsilon$, i.e., $\frac{1}{\epsilon^2}$, which accounts for the worst-case scenario.

---

### Official Review · Reviewer_5wme · 2022-07-11

**Rating:** 7
**Confidence:** 3
**Soundness:** 4 excellent
**Presentation:** 4 excellent
**Contribution:** 3 good

**Summary:**

This paper proposes the Probe-Rank algorithm for exact ranking under the Weak Stochastic Transitivity assumption. For true tranking recovery problem, the proposed algorithm guarantees instance-wise upper bound on its sample complexity. This instance-wise upper bound only depends on the preference probabilities between items that are adjacent in the true ranking. This is an improvement upon existing sample complexity results that depends on the preference probabilities for all pairs of items. Experimental results show that the proposed Probe-Rank algorithm outperforms the IIR algorithm, which is state-of-the-art in the area. In addition, a constructive example is made to illustrate the advantage of the instance-wise bound in the worst case.

**Questions:**

In Line 148, what would be the impact if the Partial order-preserving graph is not implemented?
If the proposed algorithm is implemented without Partial order-preserving graph to guarantee the acyclic/transitive property, would it become a competing method with a different line of research that does not assume coherent ranking, i.e., methods derived from Borda score and Copeland score?

In some other research, techniques to resolve the conflicts in non-coherent ranking is a practical problem. It is not in the scope of this research I believe but an interesting direction to look at if the proposed method is examined with noisy and non-coherent pairwise comparison data, which unfortunately is the case for many real-world datasets.

**Limitations:**

Given the theoretical focus, there is limited evidence of how the algorithm works on real-world data. This is acceptable but still a limitation given the presentation and length of the content.

**Strengths And Weaknesses:**

Strength:
- the writing is punchy with clear contributions and highlights
- the operational details are noted with clarity; giving readers strong confidence to reproduce the numerical results

Weakness:
- given the theoretical nature of the major contribution; the experiment focuses only on the verification of theoretical findings and lacks practical guidance on real-world data practices

---

> ### Author Response · Authors · 2022-08-02
> **Response to Reviewer 5wme**
>
> ## Reviewer 5wme
>
> We thank the reviewer for the positive feedback and appreciate the perspective about applications where rankings are non-coherent. We address the comments in detail as follows. Changes that we made in the manuscript are marked in blue.
>
> ---
> **Q1:** "given the theoretical nature of the major contribution; the experiment focuses only on the verification of theoretical findings and lacks practical guidance on real-world data practices"
>
> **A1:** We agree with the reviewer that evaluating the proposed algorithm on real-world datasets would improve the paper. We did not provide such an experiment as it can be hard to verify the WST condition and to determine the true ranking for real-world datasets.
>
> We demonstrate here, however, a small experiment derived from real-world data. We hope to include a more comprehensive experiment on real-world datasets in the future.
>
> We used the "Country Population" dataset collected from \[Jin2020\]. The dataset contains 3184 pairwise comparisons of the population size of 15 countries. Every query is made between two randomly chosen countries and the annotator chooses the country they think has a larger population. Since our algorithm and the baseline IIR algorithm both perform active sampling and the number of samples needed is random, we use this dataset to simulate queries and responses. In this experiment, we set confidence level $\delta = 0.1$ as in our main paper.
>
> We first consider a parametric approach by assuming a standard BTL model over the countries, i.e., the probability of country $i$ being perceived as having a larger population than country $j$ is $\frac{\exp(s_i)}{\exp(s_i) + \exp(s_j)}$, with $s_i,s_j$ being scores as parameters. We used the algorithm proposed in \[Jin2020\] to estimate the parameters $s_i$. Query responses are thus generated according to the BTL model with estimated parameters. Note that the BTL model satisfies the SST condition.
>
> The numbers of queries needed by the different algorithms are shown in the following tables.
>
> ``Table: Scores multiplied by 10``
>
> IIR   |  Probe-Rank |  Probe-Rank-SE
> ---- | ---- | ----
> 277000 |      33770     |    20800
>
> ``Table: Scores not scaled``
> IIR   |  Probe-Rank |  Probe-Rank-SE
> ---- | ---- | ----
>    18065985   |    7860288   | 4342208
>
>
> (Note that multiplying scores by 10 makes comparison probabilities, i.e., $\frac{\exp(s_i)}{\exp(s_i) + \exp(s_j)}$, less close to $0.5$ so that the comparisons are more accurate. In the unscaled case, there exist pairs of countries in the dataset that received the same amount of votes when being compared with each other, so the comparison probabilities under the BTL model are very close to $0.5$, thus resulting in high sample complexity.)
>
> We also considered a more direct WST setting. For every pair of items $i,j$, we approximate $P_{ij} = P(i \succ j)$ simply by the empirical winning rate, i.e., the number of times $i$ is preferred over $j$ divided by the total number of queries comparing $i$ and $j$. However, this results in a comparison matrix that does not satisfy the WST condition and some entries equal $0.5$, i.e., $P_{ij} = 0.5$ for some $i,j$. Therefore, among the 15 countries, we choose a subset of 9 by manually removing countries that violate the WST condition or with $0.5$ entries.
>
> The number of queries needed by different algorithms are shown in the following table.
>
> ``Table: WST with 9 items``
> IIR   |  Probe-Rank |  Probe-Rank-SE
> ---- | ---- | ----
>    27100   |    15960    |      7712
>
> As we can see from the above three tables, when the number of items to be ranked is small, our Probe-Rank algorithm has an advantage over the IIR algorithm. This is also suggested by our simulated experiments presented in the main paper.
>
> ---
> **Q2:** "What if the partial order-preserving graph is not implemented? If not implemented, would the proposed algorithm become a competing method with methods derived from Borda/Copeland scores? ...an interesting direction to look at if the proposed method is examined with noisy and non-coherent pairwise comparison data..."
>
> **A2:** We thank the reviewer for this comment. Indeed, the transitivity property is an assumption that we made and our algorithm highly relies on it. The partial order-preserving graph is partly used to guarantee that comparison results are well defined, i.e., we do not obtain results like $1\succ 2, 2\succ 3 ,3\succ 1$. Without the graph, our algorithm might report error or might still recover the ranking but end up performing many unnecessary comparisons. The instance-wise complexity upper bound also does not hold any more. So in applications where rankings are defined by Borda/Copeland scores, unfortunately, our algorithm is not a competing method.
>
> We also agree with the reviewer that it is a good idea to examine our algorithm in scenarios where transitivities do not hold. As in the response to **Q1**, we leave conducting a comprehensive experiment on real-world datasets to future work.

---

### Meta-Review · Area_Chair_j1pZ · 2022-08-27

**Recommendation:** Accept
**Confidence:** Certain

**Metareview:**

This is an interesting and solid paper in which the authors address the identification of a ranking in the setting of duelling bandits under the assumption that the underlying preference probabilities are weakly stochastic transitive. They introduce a novel algorithm which solves the problem in a delta-PAC manner and has an instance-wise sample complexity guarantee. All reviewers agree that this is a significant contribution. Questions and open points could be clarified in the discussion phase.

**Award:**

No

---

### Decision · Program_Chairs · 2022-09-14

Accept